# MS-BART: Unified Modeling of Mass Spectra and Molecules for Structure Elucidation

**Yang Han[1,2], Pengyu Wang[1,2], Kai Yu[1,2,4], Xin Chen[2*], Lu Chen[1,2,3,4]***

[1]X-LANCE Lab, School of Computer Science
MoE Key Lab of Artificial Intelligence, SJTU AI Institute
Shanghai Jiao Tong University, Shanghai, China
[2]Suzhou Laboratory, Suzhou, China
[3]Shanghai Innovation Institute, Shanghai, China
[4]Jiangsu Key Lab of Language Computing, Suzhou, China
{csyanghan,chenlusz}@sjtu.edu.cn,mail.xinchen@gmail.com

## Abstract

Mass spectrometry (MS) plays a critical role in molecular identification, significantly advancing scientific discovery. However, structure elucidation from MS data remains challenging due to the scarcity of annotated spectra. While large-scale pretraining has proven effective in addressing data scarcity in other domains, applying this paradigm to mass spectrometry is hindered by the complexity and heterogeneity of raw spectral signals. To address this, we propose MS-BART, a unified modeling framework that maps mass spectra and molecular structures into a shared token vocabulary, enabling cross-modal learning through large-scale pretraining on reliably computed fingerprint–molecule datasets. Multi-task pretraining objectives further enhance MS-BART's generalization by jointly optimizing denoising and translation task. The pretrained model is subsequently transferred to experimental spectra through finetuning on fingerprint predictions generated with MIST, a pre-trained spectral inference model, thereby enhancing robustness to real-world spectral variability. While finetuning alleviates the distributional difference, MS-BART still suffers molecular hallucination and requires further alignment. We therefore introduce a chemical feedback mechanism that guides the model toward generating molecules closer to the reference structure. Extensive evaluations demonstrate that MS-BART achieves SOTA performance across 5/12 key metrics on MassSpecGym and NPLIB1 and is faster by one order of magnitude than competing diffusion-based methods, while comprehensive ablation studies systematically validate the model's effectiveness and robustness. We provide the data and code at https://github.com/OpenDFM/MS-BART.

## 1 Introduction

Mass spectrometry is an analytical technique that measures the mass-to-charge ratio of ions, enabling the identification, quantification, and structural characterization of molecules. The identification of small molecules from mass spectrometry data represents a fundamental task in analytical chemistry, with broad applications across multiple domains, including drug discovery [1, 32, 39], environmental biochemistry [35], and materials science [21]. Recent advances in machine learning have enabled structure elucidation from mass spectra. Existing approaches can be broadly categorized into (1) retrieval-based methods and (2) *de novo* generative methods. Retrieval-based methods rely on matching query spectra against large annotated spectra databases. However, annotated experimental

---

* Xin Chen and Lu Chen are the corresponding authors.

39th Conference on Neural Information Processing Systems (NeurIPS 2025).

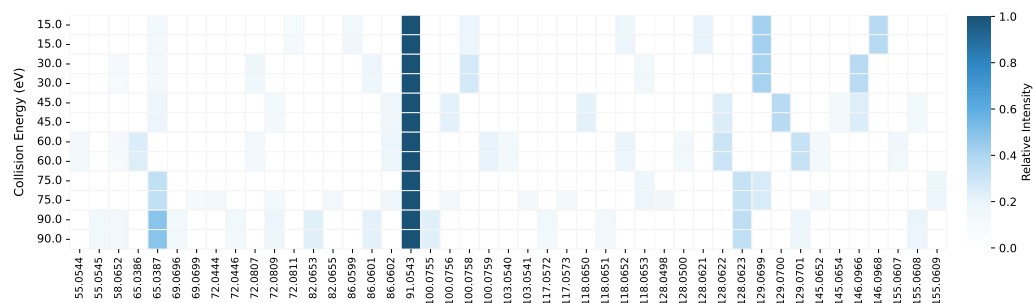

Figure 1: Randomly selected mass spectra of a molecule (SMILES: C#CCNCC1=CC=CC=C1, InChIKey: LDYBFSGEBHSTOQ) from MassSpecGym [7], acquired under varying collision energies. The $x$-axis shows the mass-to-charge ratio ($m/z$), the $y$-axis indicates collision energy (in eV), and color represents normalized relative intensity.

spectra are scarce and costly to obtain [3]. In addition, these methods are inherently limited to known molecules and cannot identify novel structures absent from reference databases. *De novo* generation methods learn to generate molecular structures directly from mass spectra, offering the potential to discover new compounds. However, these generative models [39, 45, 13, 29] are also bottlenecked by the limited availability of high-quality experimental spectra.

In other domains such as natural language processing (NLP) and computer vision (CV), a common strategy to overcome data scarcity is to pretrain models on large-scale unlabeled data and then finetune them on task-specific datasets [8, 28, 6, 20, 19]. A similar paradigm can also be observed in nuclear magnetic resonance (NMR) structure elucidation [48], where the model is first pretrained on 3.6 million unlabeled molecules and then fine-tuned directly on simulated and experimental NMR data. However, adapting this paradigm to mass spectrometry remains challenging due to the intrinsic complexity and heterogeneity of mass spectra. As illustrated in Fig. 1, spectra for the same molecule can vary substantially under different collision energies, adduct types, or instrument settings, and may even fluctuate slightly under identical experimental conditions. To address this variability, we propose using molecular fingerprints as an intermediate representation of mass spectra. Fingerprints are binary vectors that encode the presence of chemical substructures. Unlike raw spectra, they are invariant to experimental conditions and can be reliably computed from molecular structures using cheminformatics toolkits such as RDKit. This eliminates the need to simulate mass spectra under diverse experimental settings[34, 17], enabling scalable pretraining dataset construction.

Building on this insight, we propose MS-BART, a unified framework for molecular structure elucidation from mass spectrometry data, following the pretraining–finetuning–alignment paradigm widely used in NLP. We first construct a large-scale pretraining dataset consisting of fingerprint–molecule pairs, where molecular fingerprints are computed for 4 million unlabeled molecules using RDKit. Based on this dataset, we design multi-task pretraining objectives to facilitate cross-modal learning between molecular fingerprints and molecular structures. For finetuning, we incorporate experimental mass spectrometry data by using MIST [18] to predict fingerprints from spectra, conditioned on associated metadata and molecular formulas. As these predicted fingerprints are subject to dataset-specific noise and systematic biases, we finetune the pretrained model using the predicted fingerprints as input, thereby improving its adaptability to real-world experimental conditions. Finally, the generated structures are prone to molecular hallucinations [15], where outputs are chemically valid but deviate from the true molecules. To address this, we introduce an alignment step that incorporates chemical feedback by assigning higher probabilities to structures more similar to the ground truth. In summary, our main contributions are as follows:

- To the best of our knowledge, we are the first to leverage language model for mass spectra structure elucidation by introducing a unified vocabulary and multi-task pretraining on a large corpus of fingerprint–molecule pairs.

- We finetune the model on experimental data and incorporate chemical feedback to align the generative distribution with real-world structural preferences.

- We validate our approach on two public benchmarks, achieving SOTA performance across 5/12 key metrics on MassSpecGym[7] and NPLIB1[10] and is faster by one order of magnitude than competing diffusion-based methods.

## 2   Related Work

**Mass Spectra Modeling.**   Mass spectra are variable-length, discrete, two-dimensional data, which makes them inherently challenging to model. A basic approach involves padding the two-dimensional matrix to a fixed length and projecting it into an embedding space via a linear layer [45]. A more common strategy is spectral binning, which partitions spectra into fixed-width intervals (e.g., 0.1 Da) to yield fixed-size input vectors [38]. For example, ChemEmbed [14] limits molecular weights to 700 Da, encodes spectra into a 7000-dimensional vector using a bin size of 0.01, and applies a convolutional neural network (CNN) to predict 300-dimensional Mol2vec embeddings. Similarly, Spec2Mol [29] and MS2DeepScore [23] convert spectra into bit vectors and train 1D CNNs or Siamese networks [5] to learn meaningful spectral embeddings. While binning is simple, it suffers from sparsity and sensitivity to noise, limiting its ability to capture chemically meaningful patterns. To overcome this, some methods leverage molecular fingerprints as intermediate representations that encode chemical substructures more robustly. CSI:FingerID [11] predicts molecular fingerprints from tandem mass spectra using fragmentation trees and machine learning, achieving strong performance in metabolite identification. MSNovelist [41] builds on this by integrating predicted fingerprints into an encoder-decoder model for de novo structure generation. Inspired by these approaches, MS-BART adopts molecular fingerprints as a spectrum representation, enabling scalable pretraining while preserving chemical semantics.

**Structure Elucidation from Mass Spectra.**   Two major paradigms dominate structure elucidation from mass spectra: library matching and *de novo* generation. Library matching formulates the task as an information retrieval problem [40], comparing query spectra against databases of experimental or simulated spectra. Methods such as Spec2Vec [22] and MSBERT [49] learn spectrum embeddings and perform retrieval over databases like GNPS [44]. Due to the scarcity of experimental spectra, some methods (e.g., CFM-ID [43], GrAFF-MS [34]) simulate spectra from known molecular databases (e.g., PubChem). However, the effectiveness of library matching is constrained by database coverage, spectrum quality, and experimental variation, limiting its utility for novel compounds. In contrast, *de novo* approaches generate molecular structures directly from spectra, bypassing the need for reference databases. Spec2Mol [29] draws inspiration from speech-to-text models, employing an encoder-decoder network to translate spectra into SMILES sequences. MADGEN [45] uses a two-stage framework: scaffold retrieval and scaffold-conditioned molecule generation. Spectra and scaffolds are embedded into a shared latent space using MLPs and GNNs, and RetroBridge [24] generates the final structure conditioned on both inputs. Other models, including MSNovelist [41] and MS2SMILES [30], use fingerprints predicted by SIRIUS [9] as inputs to sequence models for SMILES generation. MS2SMILES further improves atom-level resolution by jointly predicting heavy atoms and their associated hydrogens. DiffMS [4] adopts an implicit fingerprint representation by extracting the final embedding from the precursor peak in MIST [18] and then generates the target structure by discrete diffusion conditioned on the spectrum embedding and node features (derived from the given formula). Despite promising results, most existing methods treat spectra and molecular structures as separate modalities, which often leads to semantic mismatches and molecular hallucinations [15]. In contrast, MS-BART unifies their modeling through a shared vocabulary. By pretraining on large-scale spectral fingerprint–molecule pairs, MS-BART learns rich representations for both chemical structures and their spectral abstractions. Subsequent finetuning and alignment on experimental spectra further enhance the model's ability to generate accurate and chemically consistent predictions on real-world data.

## 3   Methodology

Our framework is illustrated in Fig. 2. Section 3.1 introduces a unified vocabulary for representing both mass spectra and molecules. Section 3.2 describes multi-task pretraining with reliably computed fingerprints. Section 3.3 finetunes the model on experimental spectra to adapt to real-world distributions, while Section 3.4 further aligns molecular generation through chemical feedback.

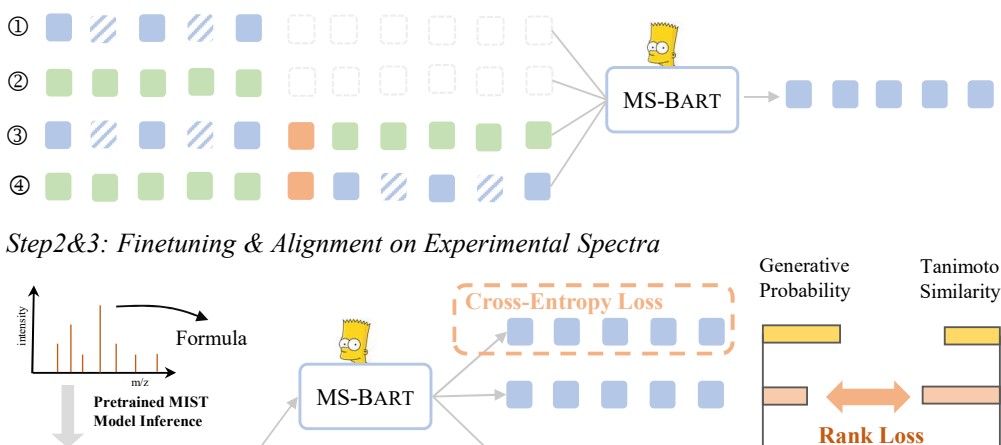

*Step1: Unified Multi-Task Pretraining on Reliably Computed Fingerprints*

*Step2&3: Finetuning & Alignment on Experimental Spectra*

Figure 2: Overview of the MS-BART framework. Square symbols represent unified tokens, where ■ denotes SELFIES tokens, ■ indicates fingerprint tokens, and ■ represents a special separator token between SELFIES and fingerprint tokens. The □ symbol signifies padding tokens. Masked tokens are represented by striped patterns (▨). The top row illustrates pretraining tasks utilizing reliably computed fingerprints (Computed by RDKit), designed to enable MS-BART to capture fundamental patterns in both fingerprint and SELFIES representations. Transfer learning is subsequently applied to experimental spectra through finetuning and alignment to enhance structure elucidation performance.

## 3.1 Mass Spectra and Molecular Representation

**Mass Spectra Representation.** Raw mass spectra consist of variable-length sets of peaks $\{P_1, P_2, \ldots, P_k\}$, where each peak $P_i = (M_i, I_i)$ represents a mass-to-charge ratio ($m/z$) and a corresponding intensity. Due to noise and variability in experimental spectra, learning from raw spectra directly is challenging. In tandem mass spectrometry, *precursor ions* undergo collision-induced dissociation (CID), producing charged *product ions* and uncharged *neutral loss* fragments. These fragmentation patterns reflect structural information and correspond to chemical fragment encoded in molecular fingerprint bits. Prior work [11, 18, 2] has explored on predicting molecular fingerprints from spectra by leveraging fragmentation information. Following this, we represent each spectrum as a 4096-bit circular Morgan fingerprint $FP \in \{0, 1\}^{4096}$, where each bit indicates the presence of a specific substructure. For experimental data, we employ the pretrained MIST model [18] to predict molecular fingerprints $FP$ from mass spectra, conditioned on the chemical formula. The output of MIST is a probability vector $\{p_0, \ldots, p_{4095}\}$, where each $p_i$ indicates the predicted likelihood of the $i$-th fingerprint bit being active. To convert these probabilities into a binary fingerprint representation, we apply a threshold $\epsilon$:

$$FP_i = \begin{cases} 1, & \text{if } p_i \geq \epsilon, \\ 0, & \text{otherwise,} \end{cases} \quad \forall i \in \{0, 1, \ldots, 4095\}. \tag{1}$$

Each activated bit ($FP_i = 1$) is converted into a fingerprint token of the form `<fp{i:04d}>` (e.g., `<fp0123>`), producing a token sequence suitable for language modeling. We also compute the fingerprints from unlabeled molecules using RDKit [27] and apply the same tokenization for subsequent pretraining.

**Molecular Representation.** SMILES [46] and SELFIES [25] are two widely used string-based molecular representations. While SMILES is compact and human-readable, it does not guarantee chemical validity. In contrast, SELFIES is designed to ensure that every valid string maps to a chemically feasible molecule. We adopt SELFIES in MS-BART for its robustness and validity guarantees. To ensure consistency and uniqueness, we use the canonical form of each SELFIES string. Following [15], we employ a vocabulary of 185 SELFIES tokens. Although this is significantly smaller than typical language model vocabularies, prior work [37] shows that compact chemical vocabularies are sufficient for effective molecular representation learning.

## 3.2 Unified Multi-Task Pretraining on Reliably Computed Fingerprints

Given the tokenization approach described above, we tokenize mass spectra and molecular sequences by MIST [18] and the aforementioned vocabulary, respectively. As illustrated in Fig. 2, we design three self-supervised denoising tasks for pretraining to recover masked spans, along with one cross-modal translation task to strengthen modality alignment. We pretrain MS-BART on a simulated dataset of 4 million fingerprint–molecule pairs. The molecules are provided by MassSpecGym [7] and are preprocessed by excluding those with an MCES distance of less than two from any molecule in the test fold. The pretraining framework comprises four tasks detailed as follows: (1) *SELFIES Denoising (①).* Randomly mask 30% of tokens in SELFIES sequence $S = \{s_1, \dots, s_l\}$ with $\texttt{[MASK]}$ token and recover original tokens. (2) *Fingerprint-to-Molecule Translation (②).* Generate SELFIES sequences conditioned on fingerprint tokens $FP$. (3) *Hybrid Denoising (③ and ④).* Combine fingerprint tokens and masked SELFIES using separator $\texttt{<fps\_sep>}$, with input order variants, $[FP, \texttt{<fps\_sep>}, S_{masked}]$ and $[S_{masked}, \texttt{<fps\_sep>}, FP]$, predicting full SELFIES sequences from both modalities. All tasks follow a conditional generation paradigm optimized via cross-entropy loss:

$$\mathcal{L}_{ce} = -\sum_{i=1}^{l} \log P(y_i \mid y_{<i}, X; \theta), \tag{2}$$

where $\theta$ denotes the model parameters, $X$ is the input sequence (e.g., masked SELFIES or fingerprints), $y_i$ is the $i$-th target token, and $l$ is the target SELFIES length.

## 3.3 Finetuning on Experimental Spectra

After pretraining on the simulated fingerprint-molecule dataset, MS-BART is fine-tuned on experimental spectra to bridge the domain gap between computational and real-world data distributions. As shown in Fig. 2, the original mass spectra are tokenized into fingerprint tokens, and the model is also optimized through cross-entropy loss (Eq. 2) calculated between the target and predicted SELFIES tokens. This crucial step aims to learn the systematic bias introduced by the MIST [18] model and improve prediction performance.

## 3.4 Contrastive Alignment via Chemical Feedback

Following pretraining and dataset-specific fine-tuning, MS-BART acquires the capability to interpret molecular fingerprints and generate chemically plausible molecular structures. However, it remains susceptible to *molecular hallucination* [15], where generated molecules maintain chemical validity but exhibit limited consistency with original mass spectra or corresponding fingerprints. Specifically, the model may yield structures deviating substantially from true underlying molecular structures. To alleviate this hallucination and enhance performance, we propose aligning the model's probabilistic rankings of generated molecules with preference rankings derived from chemical contexts. In this paper, we define molecular preference through Tanimoto similarity, denoted as $Ps(\cdot)$. Given a fingerprint $FP$, MS-BART generates $n$ candidate molecules $C = \{S_1, S_2, \cdots, S_n\}$. The preference score for each candidate $S_i$ is calculated as $Ps(S_i) = Tan(S_i, S)$, where $S$ represents the ground-truth molecular structure. Simultaneously, the model (parameterized by $\theta$) assigns a conditional log-probability estimate $P_\theta(S_i)$ to each candidate $S_i$, given the input $FP$. Our objective is to establish consistency between the model's generative probabilities and Tanimoto similarity metrics. Specifically, for any candidate pair $(S_i, S_j)$, we expect:

$$P_\theta(S_i) > P_\theta(S_j), if\ Ps(S_i) > Ps(S_j). \tag{3}$$

To encourage MS-BART to assign higher probabilities to candidate molecules that are more structurally similar to the target molecule, we employ a contrastive rank loss [31, 15], defined as:

$$\mathcal{L}_{\text{rank}}(C) = \sum_{i} \sum_{j>i} \max\left(0, P_\theta(S_j) - P_\theta(S_i) + \gamma_{ij}\right), \quad \forall i < j,\ \text{Ps}(S_i) > \text{Ps}(S_j), \tag{4}$$

where $\gamma_{ij} = (j - i) * \gamma$ denotes a margin scaled by the rank difference between candidates, $\gamma$ is a hyperparameter. Additionally, we retain the token-level cross-entropy loss (Eq. 2) to preserve the

model's generative capability and the overall loss is defined as:

$$\mathcal{L} = \mathcal{L}_{ce} + \alpha \mathcal{L}_{\text{rank}}, \tag{5}$$

where $\alpha$ controls the weight of the rank loss. By jointly optimizing the token-level cross-entropy loss and the sequence-level contrastive rank loss on the same finetuning dataset, MS-BART can assign a balanced probability mass across the whole sequence. This optimization strategy elevates the probability of generating molecular structures that not only exhibit higher similarity to the target molecule but may also achieve exact matches.

## 4  Experiments

### 4.1  Datasets

We evaluate our MS-BART model on two widely used open-source benchmarks: NPLIB1 [10] and MassSpecGym [7], following prior works [4, 45]. NPLIB1 is a subset of the GNPS library, originally curated for training the CANOPUS model. Its name serves to distinguish the dataset from the associated tool. MassSpecGym is the largest publicly available dataset containing 231k high-quality mass spectra spanning 29k unique molecular structures. The dataset is partitioned into training, validation, and test sets based on the edit distance between molecular structures, facilitating robust evaluation. Although MADGEN [45] also reports performance on the NIST23 dataset, access to this resource is restricted due to its commercial licensing requirements.

### 4.2  Evaluation Metrics and Baselines

To evaluate the performance of our model, we employ the following metrics:

- **Top-$k$ accuracy:** We measure the exact match between the predicted structure and the ground truth molecule by converting the generated molecule into a full InChIKey and comparing it with the gold InChIKey. Since MS fragmentation is largely insensitive to 3D stereochemistry, results based on 2D InChIKey are also reported in Appendix C.
- **Top-$k$ maximum Tanimoto similarity:** This metric quantifies the structural similarity between molecules using molecular fingerprints. We compute fingerprints based on the Morgan algorithm [33] with a radius of 2 and a bit length of 2048 using RDKit.
- **Top-$k$ minimum MCES (maximum common edge subgraph):** This metric measures the graph edit distance between molecules, reflecting the largest common substructure shared between the generated and ground-truth molecules [26].

We report the $k = 1, 10$ metrics following previous works [4, 7, 45]. Meanwhile, DIFFMS samples 100 molecules for each spectrum and identifies the top-$k$ molecules based on frequency. To ensure a fair comparison, we sample 100 molecules and subsequently rank the generated molecules according to their distance from the given formula. Given two formula $F_1 = \{(a_1, n_1), (a_2, n_2), ..., (a_m, n_m)\}$ and $F_2 = \{(a_1, m_1), (a_2, m_2), ..., (a_m, m_k)\}$, the distance is defined as:

$$D(F_1, F_2) = \sum_{a \in \text{All Atoms}} |n_a - m_a|, \tag{6}$$

$n_a, m_a$ are the counts of atom $a$ in $F_1$ and $F_2$. If multiple molecules have the same distance, we sort them according to their estimated log-probability. After this re-ranking, we select the first $k$ molecules as the top-$k$ predictions.

**Baselines.**  MassSpecGym [7] establishes three baselines for molecular generation: random generation, a SMILES-based Transformer, and a SELFIES-based Transformer. Spec2Mol [29] is retrained on both the NPLIB1 and MassSpecGym datasets to enable fair comparison. MIST+MSNovelist modifies the original MSNovelist framework [41] by replacing CSI:FingerID [11] with MIST. In MIST+Neuraldecipher, molecules are encoded into CDDD representations [47], followed by reconstruction of the SMILES strings using a pretrained LSTM decoder. MADGEN [45] and DIFFMS [4] represent recent state-of-the-art approaches. MADGEN first retrieves molecular scaffolds, then generates complete structures using the RetroBridge model [24], conditioned on both spectra and scaffolds. DIFFMS also adopts MIST as the spectrum encoder and employs a Graph Transformer [12] as the diffusion decoder, with separate pretraining of encoder and decoder components.

Table 1: Performance comparison of MS-BART and baseline methods on the NPLIB1 [10] and MassSpecGym [7]. Results marked with * are reproduced from MassSpecGym and DIFFMS. **Bold** denotes the best performance, underlined indicates the second-best.

| Model | Top-1 | | | Top-10 | | |
|---|---|---|---|---|---|---|
| | Accuracy ↑ | MCES ↓ | Tanimoto ↑ | Accuracy ↑ | MCES ↓ | Tanimoto ↑ |
| **NPLIB1** | | | | | | |
| Spec2Mol* | 0.00% | 27.82 | 0.12 | 0.00% | 23.13 | 0.16 |
| MIST + Neuraldecipher* | 2.32% | 12.11 | 0.35 | 6.11% | 9.91 | 0.43 |
| MIST + MSNovelist* | 5.40% | 14.52 | 0.34 | 11.04% | 10.23 | 0.44 |
| MADGEN | 2.10% | 20.56 | 0.22 | 2.39% | 12.69 | 0.27 |
| DIFFMS | **8.34%** | 11.95 | 0.35 | **15.44%** | 9.23 | 0.47 |
| MS-BART | 7.45% | **9.66** | **0.44** | 10.99% | **8.31** | **0.51** |
| MS-BART(Gold Fingerprint) | 73.50% | 2.14 | 0.90 | 79.12% | 1.60 | 0.94 |
| **MASSSPECGYM** | | | | | | |
| SMILES Transformer* | 0.00% | 79.39 | 0.03 | 0.00% | 52.13 | 0.10 |
| SELFIES Transformer* | 0.00% | 38.88 | 0.08 | 0.00% | 26.87 | 0.13 |
| Random Generation* | 0.00% | 21.11 | 0.08 | 0.00% | 18.26 | 0.11 |
| Spec2Mol* | 0.00% | 37.76 | 0.12 | 0.00% | 29.40 | 0.16 |
| MIST + Neuraldecipher* | 0.00% | 33.19 | 0.14 | 0.00% | 31.89 | 0.16 |
| MIST + MSNovelist* | 0.00% | 45.55 | 0.06 | 0.00% | 30.13 | 0.15 |
| MADGEN | 1.31% | 27.47 | 0.20 | 1.54% | 16.84 | 0.26 |
| DIFFMS | **2.30%** | 18.45 | **0.28** | **4.25%** | **14.73** | **0.39** |
| MS-BART | 1.07% | **16.47** | 0.23 | 1.11% | 15.12 | 0.28 |
| MS-BART(Gold Fingerprint) | 47.56% | 3.26 | 0.85 | 64.62% | 2.02 | 0.93 |

## 4.3 Implementation

We adopt BART-BASE [28] as the backbone of MS-BART, initializing all parameters from scratch using a normal distribution. To convert MIST probabilities into binary fingerprints, we apply a threshold of $\epsilon = 0.2$ for NPLIB1 and $\epsilon = 0.11$ for MassSpecGym. Further details regarding this selection are provided in Appendix A. During pretraining, we set the maximum sequence length to 512 to accommodate simultaneous learning from both fingerprints and SELFIES. For finetuning and alignment, we fix the input and output token lengths to 256, as the fingerprint inputs and SELFIES outputs rarely exceed this length in practice. When aligning MS-BART with chemical feedback, we freeze the encoder following practices from prior work [16, 36], and update only the decoder. Additional training details are provided in Appendix B.

## 4.4 Main Results

Table 1 presents a comparison of overall performance, demonstrating that MS-BART outperforms most baseline methods and achieves SOTA performance across 5/12 key metrics on NPLIB1 and MassSpecGym. Notably, on NPLIB1, MS-BART performs the best across all similarity metrics, surpassing the second-best method by 19.16% (MCES) and 25.71% (Tanimoto similarity) in the Top-1 setting. Significant improvements are also observed in the Top-10 setting, further validating the effectiveness of MS-BART. The high absolute Tanimoto similarity values (close to or exceeding 0.5) suggest that MS-BART generates structurally similar molecules, which are particularly valuable to domain experts. However, the Top-1 and Top-10 accuracy do not surpass DiffMS [4], primarily because we have filtered out similar molecules in the pretraining data (Appendix B), whereas DiffMS only removes all NPLIB1 and MassSpecGym test and validation molecules from their pretraining dataset. Another recently proposed open-source dataset, MassSpecGym, presents a more challenging benchmark than NPLIB1, as NPLIB1 lacks a scaffold-based split, resulting in a test set containing molecules with high structural similarity (Tanimoto similarity > 0.85) to those in the training set [4, 7]. Meanwhile, MassSpecGym exhibits a more complex data composition due to the presence of $[M+Na]^+$ adducts. The $Na^+$ atom have a higher mass than $H^+$ atom, leading to more complex fragmentation patterns and consequently a data distribution that differs significantly from that of $[M+H]^+$. Furthermore, the $[M+Na]^+$ data in the MassSpecGym training set is relatively scarce, constituting only 15.52% of the total samples. This class imbalance, combined with the distinct fragmentation behavior of $[M+Na]^+$ compared to $[M+H]^+$, introduces additional noise and potential bias during model training. To avoid fragmentation pattern conflicts and preserve data consistency, we

Table 2: Performance comparison of MS-BART on the NPLIB1 dataset using different pretraining strategies. "NONE" indicates no pretraining, "SD" and "TRANS" denote pretraining with *SELFIES Denoising* and *Fingerprint-to-Molecule Translation*, respectively, and "HYBRID" refers to pretraining with the *Hybrid Denoising* method described in Section 3.2.

| PRETRAIN STRATEGY | TOP-1 | | | TOP-10 | | |
|---|---|---|---|---|---|---|
| | ACCURACY ↑ | MCES ↓ | TANIMOTO ↑ | ACCURACY ↑ | MCES ↓ | TANIMOTO ↑ |
| NONE | 1.71% | 12.93 | 0.27 | 3.05% | 11.36 | 0.34 |
| SD | 0.37% | 14.41 | 0.24 | 0.98% | 12.42 | 0.32 |
| TRANS | 6.23% | **9.37** | 0.42 | 10.26% | **7.98** | 0.50 |
| HYBRID | 5.13% | 9.96 | 0.41 | 7.81% | 8.87 | 0.48 |
| MS-BART | **7.45%** | 9.66 | **0.44** | **10.99%** | 8.31 | **0.51** |

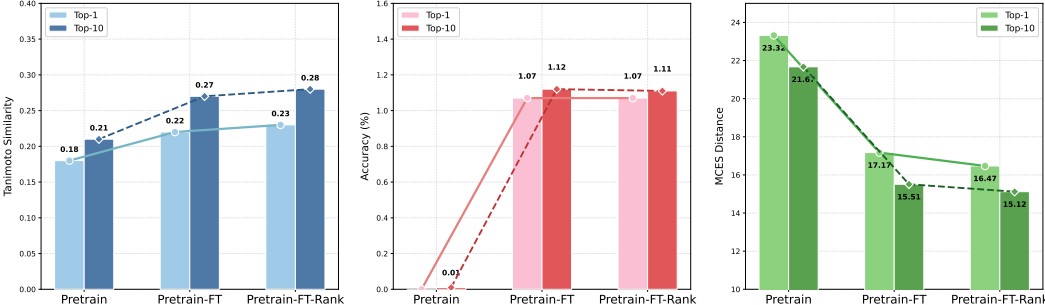

Figure 3: Progressive improvement of MS-BART for molecular hallucination mitigation on MassSpec-Gym. Subfigures show: a) Tanimoto similarity, b) Accuracy, and c) MCES scores across three training stages under Top-1 and Top-10 settings. Stage 1: Pretrain, the base model pretrained on 4M simulated unlabeld molecules. Stage 2: Pretrain-FT, fine-tuned on MassSpecGym. Stage 3: Pretrain-FT-Rank, fully optimized MS-BART with chemical feedback.

filter out $[M+Na]^+$ adducts before fine-tuning and alignment, retaining only the dominant $[M+H]^+$ data. However, to ensure a fair and comprehensive evaluation, we retain the $[M+Na]^+$ adducts in the test set. The results show that MS-BART does not surpass DiffMS on most metrics, and the main reason is same as for NPLIB1. Excluding DiffMS, MS-BART also shows SOTA performance across all similarity metrics and demonstrates the robustness and effectiveness of MS-BART in handling diverse spectral patterns. Finally, we report the best possible performance of MS-BART if we use the gold fingerprint calculated from the true structure rather than predicting it with the pretrained MIST model. It is noteworthy that MS-BART can almost find the exact match or extremely similar candidates, proving the great potential of MS-BART and indicating that further work can be devoted to improving the performance of MIST model.

## 4.5 Unified Multi-Task Pretraining Enhances Cross-Modal Learning

Pretraining serves as the foundational phase and a critical step in training language models, enabling a broad understanding of linguistic features such as syntax, semantics, and context, which are essential for effective transfer learning. To investigate the role of multi-task pretraining in enhancing MS-BART's comprehension of molecular fingerprints and SELFIES, we conduct ablation experiments, with results presented in Table 2. It is obvious that the model trained without pretraining exhibits relatively poor performance compared to its pretrained counterpart. Nevertheless, its accuracy remains above chance level and surpasses baseline methods that encode mass spectra directly, demonstrating the advantage of representing raw mass spectra as fingerprints and training with a unified vocabulary in an end-to-end style. Furthermore, we compare multi-task pretraining with single-task pretraining. Pretraining solely with the denoising task leads to performance degradation rather than improvement, primarily because the denoising task is not well aligned with structure elucidation. The substantial improvement observed in the fingerprint-to-molecule translation task further supports this finding. Moreover, the performance gain of MS-BART over single-task pretraining indicates that the denoising task remains beneficial, as it helps MS-BART develop a fundamental understanding of molecular

structures and contributes to the final performance. These results suggest that unified multi-task pretraining on unlabeled data promotes cross-modal interaction and alignment by enabling MS-BART to learn a shared representation space across modalities.

## 4.6 MS-BART Mitigates Molecular Hallucination

The training paradigm of MS-BART consists of three steps. Fig. 3 illustrates the impact of these steps on the final performance in MassSpecGym, revealing two key findings. First, continued fine-tuning on the MassSpecGym training fold improves performance and mitigates molecular hallucination, particularly in terms of accuracy. For example, the "Pretrain" model achieves near-zero Top-1 and Top-10 accuracy scores of 0.00% and 0.01%, respectively, while the "Pretrain-FT" model improves these values to 1.07% and 1.12%. Moreover, aligning the model with chemical feedback based on Tanimoto similarity further reduces molecular hallucination. This alignment enables the generation of structurally more similar molecules, as evidenced by the improved Tanimoto similarity and the degraded MCES metric. Fig. 4 shows a randomly selected case from MassSpecGym where MS-BART effectively mitigates molecular hallucination.

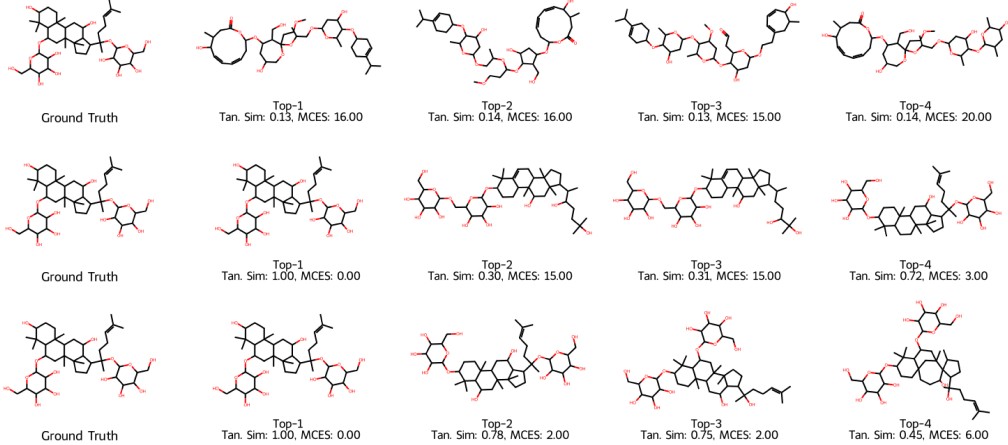

Figure 4: Predictions of the Pretrain (top row), Pretrain-FT (middle row), and Pretrain-FT-Rank (bottom row) models on a representative MassSpecGym sample, with the ground-truth structure shown in the first column and the Top-4 generated outputs in the subsequent columns. Results demonstrate that MS-BART can effectively mitigate molecular hallucination and predict more chemically plausible and structurally consistent predictions.

## 4.7 Sensitivity Analysis of Model Hyperparameters

MS-BART employs two key hyperparameters, the fingerprint generation threshold $\epsilon$ and the rank loss weight $\alpha$. The fingerprint threshold determines how the MIST probability is converted into an active fingerprint, where different thresholds lead to distinct representations. The rank loss weight controls the influence of the token-level rank loss in guiding the alignment training. To comprehensively evaluate the impact of these two parameters, we experiment with different values of $\epsilon$ on MassSpecGym and $\alpha$ on NPLIB1 and present the results in Table 3. The results show no significant differences among the tested $\epsilon$ values, indicating that MS-BART is not sensitive to this parameter. Although $\epsilon = 0.11$ achieves the best performance on the MassSpecGym validation set and is reported in Table 1 as the final result, the ablation results suggest that $\epsilon = 0.11$ is not the best overall. This discrepancy is mainly due to the high difficulty of the MassSpecGym dataset and the distribution mismatch between its validation and test sets. A similar observation holds for $\alpha$ on NPLIB1. The model with $\alpha = 5$ performs best in terms of Top-1 Tanimoto score on the validation set and is also reported as the final result. However, models with $\alpha = 1$ and $\alpha = 3$ also achieve competitive results on specific metrics. The variations across different configurations are minor, indicating that MS-BART is not highly sensitive to the rank loss weight.

## 4.8 Analysis of Decoding Hyperparameters

Table 3: Performance comparison of MS-BART under two key model hyperparameters, where the ablation of the fingerprint generation threshold and the rank loss weight are evaluated on MassSpec-Gym and NPLIB1, respectively.

| | Top-1 | | | Top-10 | | |
|---|---|---|---|---|---|---|
| | Accuracy ↑ | MCES ↓ | Tanimoto ↑ | Accuracy ↑ | MCES ↓ | Tanimoto ↑ |
| **Fingerprint Generation Threshhold $\epsilon$ (MassSpecGym)** | | | | | | |
| $\epsilon = 0.11$ | 1.07% | 16.47 | 0.23 | 1.11% | 15.12 | 0.28 |
| $\epsilon = 0.15$ | 1.20% | 16.88 | 0.23 | 1.20% | 15.45 | 0.28 |
| $\epsilon = 0.20$ | 1.19% | 17.27 | 0.23 | 1.20% | 15.75 | 0.28 |
| **Rank Loss Weight $\alpha$ (NPLIB1)** | | | | | | |
| $\alpha = 1$ | 7.08% | 10.56 | 0.44 | 12.45% | 9.10 | 0.52 |
| $\alpha = 3$ | 7.20% | 9.53 | 0.44 | 10.50% | 8.21 | 0.51 |
| $\alpha = 5$ | 7.45% | 9.66 | 0.44 | 10.99% | 8.31 | 0.51 |

MS-BART employs beam-search multinomial sampling for structure prediction, governed by two key decoding hyperparameters while maintaining default values for other parameters. First, the temperature parameter [50, 42] regulates output randomness during inference, a factor known to substantially influence generation quality in typical language models. Interestingly, MS-BART's outputs demonstrate remarkable stability across temperature variations, which is likely attributable to the model's high confidence in next-token prediction gained through multi-task pretraining and its relatively small vocabulary size of 185. Detailed temperature analysis is provided in Appendix D. The second critical parameter, beam width, governs the search space breadth during decoding. We systematically evaluated MS-BART's performance on the complete MassSpecGym test fold using an NVIDIA A800-SXM4-80GB GPU, exploring beam widths from 10 to 100. As shown in Fig. 5, both Top-1 and Top-10 accuracies demonstrate consistent improvement with increasing beam width, while inference latency exhibits linear scaling. Notably, when tested on a common consumer-grade GPU like the RTX 4090 with a beam width of 100, the average inference time per spectrum is about 3 seconds, which is 53x times faster than DiffMS's approximately 160 seconds and remains practically acceptable.

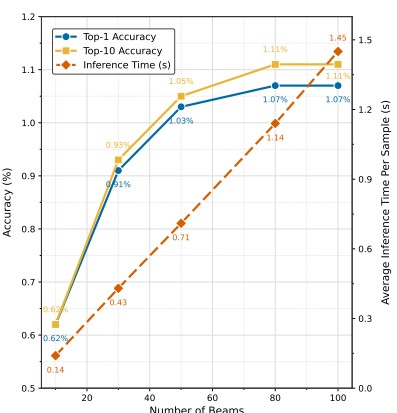

Figure 5: Performance of MS-BART with beam-search multinomial sampling decoding under different beam widths: Top-1 accuracy, Top-10 accuracy, and average inference time per spectrum.

## 5   Conclusion

In this work, we propose MS-BART, a novel language model for mass spectra structure elucidation within a unified modeling framework. Specifically, we first represent mass spectra as fingerprints and tokenize both the fingerprints and molecular representations (SELFIES) using a unified vocabulary. Subsequently, MS-BART undergoes the standard pretraining-finetuning-alignment paradigm commonly employed in NLP, enabling the model to interpret spectral fingerprints and generate plausible molecular structures. Extensive experiments demonstrate that MS-BART achieves SOTA performance on two widely adopted benchmarks across 5/12 key metrics on MassSpecGym and NPLIB1 and is faster by one order of magnitude than competing diffusion-based method, while ablation studies further validate its effectiveness and robustness.

## Acknowledgments and Disclosure of Funding

This work was supported by the National Science and Technology Major Project (2023ZD0120703), the China NSFC Projects (U23B2057, 62120106006, and 92370206), and Shanghai Municipal Science and Technology Projects (2021SHZDZX0102 and 25X010202846).

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

## A    Selection of the Probability Threshold $\epsilon$

We selected the threshold value $\epsilon$ for NPLIB1 and MassSpecGym using different methods. Since NPLIB1 is known to be a much easier dataset, we chose its threshold empirically. After examining the probability distribution, we observed that most probabilities were smaller than 0.3. A large threshold could cause information loss, whereas a small threshold might introduce noise. Therefore, we selected $\epsilon = 0.2$ to balance these factors, which resulted in excellent performance. For MassSpecGym, we trained MS-BART using $\epsilon$ values ranging from 0.1 to 0.2 in increments of 0.01. We then validated these on the validation set and selected the best threshold based on the highest Top-1 Tanimoto similarity. As shown in Table 4, the Top-1 Tanimoto similarity values for different thresholds were very close. We ultimately chose $\epsilon = 0.11$ as the final threshold, although performance did not vary significantly across the different threshold values.

Table 4: Top-1 Tanimoto Similarity under different $\epsilon$ values.

| $\epsilon$ | 0.10 | 0.11 | 0.12 | 0.13 | 0.14 | 0.15 | 0.16 | 0.17 | 0.18 | 0.19 | 0.20 |
|---|---|---|---|---|---|---|---|---|---|---|---|
| Tanimoto Similarity | 0.1666 | **0.1678** | 0.1640 | 0.1651 | 0.1660 | 0.1660 | 0.1654 | 0.1643 | 0.1651 | 0.1649 | 0.1636 |

## B    MS-BART Training Details

**Pretraining.**    We pretrain MS-BART from scratch on a simulated dataset of 4M fingerprint-molecule pairs. This dataset was generated from a refined subset of the 4M unlabeled molecules provided by MassSpecGym [7]. The refinement process involved excluding any molecule with an MCES distance of less than two from any molecule in the MassSpecGym test fold. For the NPLIB1 dataset, we first evaluated the structural similarity between its test set and our pretraining data. As shown in Table 5, approximately 3% of the pretraining molecules were structurally similar to those in the NPLIB1 test set. To prevent data leakage, we further filtered the pretraining set to remove any molecules that were structurally similar or identical to those in the NPLIB1 test set. Specifically, we refined the

pretraining dataset by removing molecules with a maximum Tanimoto similarity greater than 0.5 to any molecule in the NPLIB1 test set.

The training was conducted on four NVIDIA A800-SXM4-80GB GPUs using bfloat16 precision. A per-device batch size of 96 and two gradient accumulation steps were employed, resulting in an effective total batch size of 768 across three training epochs. The optimization process adopted a cosine learning rate scheduler with a warm-up phase of 10,000 steps. The learning rate increased linearly from zero to a peak value of 6e-4 during warm-up and subsequently decayed following a cosine schedule to a minimum value of 1e-5. The entire multi-task pretraining process required approximately 34 hours to complete.

Table 5: Distribution of max Tanimoto Similarity between pretraining set and NPLIB1 test set.

| Similarity Interval | 0.0-0.1 | 0.1-0.2 | 0.2-0.3 | 0.3-0.4 | 0.4-0.5 | 0.5-0.6 | 0.6-0.7 | 0.7-0.8 | 0.8-0.9 | 0.9-1.0 |
|---|---|---|---|---|---|---|---|---|---|---|
| Proportion | 0.09% | 8.40% | 55.85% | 21.58% | 6.79% | 3.79% | 1.84% | 1.20% | 0.40% | 0.06% |

**Finetuning.** Fine-tuning is performed on a single NVIDIA A800-SXM4-80GB GPU using bfloat16 precision, with consistent parameter settings across both MassSpecGym and NPLIB1 datasets. We adopt a learning rate of 5e-5 combined with a warm-up phase covering 10% of total training steps. Each training iteration processes 128 samples per batch. For validation monitoring, we implement an early stopping criterion that evaluates model performance every 400 steps on MassSpecGym and every 200 steps on NPLIB1. Training terminates when the validation set's Top-1 Tanimoto similarity fails to improve for three consecutive evaluations.

**Alignment Training.** The alignment phase is more complex than pretraining and fine-tuning, requiring careful hyperparameter optimization. As specified in Table 6, we explore multiple configurations during this stage while maintaining a fixed batch size of 128 and bfloat16 precision on a single NVIDIA A800 GPU. The learning schedule includes a 10% warm-up ratio of total training steps. Validation frequency is set to 400-step intervals for MassSpecGym and 50-step intervals for NPLIB1. We extend the patience to five consecutive evaluations without improvement in Top-1 Tanimoto similarity before triggering early stopping. Final hyperparameter selection is determined by maximizing the validation set's Top-1 Tanimoto similarity.

Table 6: Hyper-parameter settings.

| Hyper-parameters | Values |
|---|---|
| Learning Rate | {1e-5, 5e-5} |
| Candidate Margin $\gamma$ | {0.05, 0.1, 0.2} |
| Rank Loss Weight $\alpha$ | {1, 3, 5} |
| Number of Candidates | {3, 5} |
| Length Penalty Coefficient | {1.4, 1.6} |

## C  2D InChIKey Based Accuracy

We evaluate the Top-1 and Top-10 accuracy based on 2D InChIKey matching and present the results in Table 7. On NPLIB1, the two calculation methods yield the same results. However, for MassSpecGym, the 2D InChIKey-based accuracy of MS-BART shows a slight improvement compared to evaluation with the full InChIKey because the 2D InChIKey uses only the first 14 characters, which do not include 3D stereochemistry. To maintain consistency with the DiffMS [4], we use the full InChIKey results as the final results.

## D  The Impact of Sampling Temperature on Model Performance

Table 8 presents the performance evaluation of MS-BART during decoding with varying temperature values on MassSpecGym. The results demonstrate that MS-BART's performance remains largely consistent across different temperature settings. This stability can be attributed to the model's high confidence in next-token predictions, which likely stems from its multi-task pretraining framework.

Table 7: 2D InChIKey based Top-1 and Top-10 accuracy on NPLIB1 and MassSpecGym.

| Calculation Method | Top-1 Accuracy | Top-10 Accuracy |
|---|---|---|
| **NPLIB1** | | |
| MS-BART (InChIKey) | 7.45% | 10.99% |
| MS-BART (2D InChIKey) | 7.45% | 10.99% |
| **MASSSPECGYM** | | |
| MS-BART (InChIKey) | 1.07% | 1.11% |
| MS-BART (2D InChIKey) | 1.26% | 1.28% |

Table 8: The performance of MS-BART when decoding with different temperature on MassSpecGym.

| TEMPERATURE | TOP-1 | | | TOP-10 | | |
|---|---|---|---|---|---|---|
| | ACCURACY ↑ | MCES ↓ | TANIMOTO ↑ | ACCURACY ↑ | MCES ↓ | TANIMOTO ↑ |
| **Prtrain-FT** | | | | | | |
| 0.2 | 1.07% | 17.16 | 0.22 | 1.12% | 15.51 | 0.27 |
| 0.4 | 1.07% | 17.17 | 0.22 | 1.12% | 15.51 | 0.27 |
| 0.8 | 1.07% | 17.16 | 0.22 | 1.12% | 15.50 | 0.27 |
| **MS-BART** | | | | | | |
| 0.2 | 1.07% | 16.47 | 0.23 | 1.11% | 15.12 | 0.28 |
| 0.4 | 1.07% | 16.47 | 0.23 | 1.11% | 15.12 | 0.28 |
| 0.8 | 1.07% | 16.47 | 0.23 | 1.11% | 15.11 | 0.28 |

# E  Limitations

Our primary limitation lies in the reliance on the external MIST model [18] for predicting fingerprints from experimental spectra. The accuracy of these fingerprint predictions significantly impacts the overall performance. Future work may focus on fine-tuning the MIST model using task-specific datasets to improve prediction accuracy, or it could explore directly modeling the fragments from raw mass spectra. Another limitation is that the formula information is used only for re-ranking and not for training the model. However, these additional formulas definitely include important information and are very likely to boost the model's performance. We plan to explore effective ways to incorporate these informational hints into MS-BART in future work.

# F  Ethics Statement

All data used in this work are obtained from publicly available sources, including molecular structure datasets and mass spectrometry benchmarks such as NPLIB1 and MassSpecGym. We strictly follow the corresponding licenses and usage protocols. No modifications have been made to the original datasets beyond necessary preprocessing using standard cheminformatics tools (e.g., RDKit) to compute molecular fingerprints. No personal, private, or sensitive data are involved. The proposed model, MS-BART, is trained and evaluated solely for the task of molecular structure elucidation from mass spectrometry data. Our work does not involve human subjects, biometric data, or decision-making in socially sensitive applications. We do not foresee any immediate negative societal impact. On the contrary, improving molecular identification from mass spectrometry may benefit fields such as drug discovery, environmental chemistry, and materials science.

