# OpenReview forum: "MS-BART: Unified Modeling of Mass Spectra and Molecules for Structure Elucidation"
_NeurIPS.cc/2025/Conference — NeurIPS 2025 poster_

### Official Review · Reviewer_n2SS · 2025-06-19

**Clarity:** 3
**Significance:** 3
**Originality:** 3
**Rating:** 5
**Confidence:** 4

**Summary:**

The paper presents MS-BART — a BART-based model for predicting SELFIES-encoded molecular structures from Morgan fingerprints with the application to annotate tandem mass spectra. The method is trained in three stages:
(i) pretraining on SELFIES–fingerprint pairs from a dataset of molecular structures,
(ii) fine-tuning on a dataset of SELFIES–fingerprint pairs from a mass spectral library, where the fingerprints are predicted using a pretrained MIST model,
(iii) additional contrastive alignment based on Tanimoto similarities of sampled fingerprints to improve fingerprint prediction.

The method is overall solid and clearly presented. The authors also provide ablation studies and qualitative examples showing the benefits of multiple training stages and several pretraining tasks during stage (i). However, the experimental results have some issues.

**Questions:**

- Would MS-BART have a state-of-the-art performance under corrected evaluation protocols? My current score is 3: Borderline reject. However, if the authors address my major concerns and demonstrate that the method indeed achieves state-of-the-art results, I will raise my score to 5: Accept.
- MIST does not appear to have a particularly excellent performance on MassSpecGym. Can MS-BART effectively handle predicted fingerprints which may be low-quality?
- Did the authors experiment with adding chemical formula tokens to the input/output sequence of MS-BART?

**Ethical Concerns:**

["NO or VERY MINOR ethics concerns only"]

**Final Justification:**

My concerns regarding data leakage were addressed by the authors, I am raising my score to 5.

**Limitations:**

yes

**Paper Formatting Concerns:**

No formatting concerns

**Quality:**

2

**Strengths And Weaknesses:**

Below, I summarize the strengths of the paper, as well as, my major concerns and minor comments.
## Strengths

### Quality

The method is well structured and technically sound, with thoughtful integration of generative modeling, fingerprint conditioning, and contrastive learning. The three-stage training pipeline is clearly explained, and ablation studies support the benefit of each stage.
### Clarity

The paper overal easy to follow. The description of the model architecture and training stages is well organized.

### Significance

The task of molecular structure elucidation from tandem mass spectra is highly relevant to the metabolomics and computational chemistry communities. The proposed method addresses this problem with a combination of modern tools and training strategies that, if properly validated, could set a new performance baseline on public benchmarks.

### Originality

The work is original in its integration of BART-based decoding with fingerprint supervision and contrastive alignment. While prior models have used generative approaches and fingerprint conditioning, this work offers a unified and scalable framework that extends current techniques.

## Weaknesses

### Major concerns

- Current experimental results appear to suffer from data leakage:

    - MassSpecGym. Could the authors clarify what data was used to train the MIST fingerprint predictor or what checkpoint was used for each benchmark (MassSpecGym and NPLIB1)? If MIST was trained on NPLIB1 data (i.e., using the checkpoint from the official repository), the reported results on MassSpecGym are not valid. In this case, MIST’s training data may contain entries from the MassSpecGym test folds, since both datasets include spectra from mass spectral libraries such as MoNA and GNPS. This would mean that test results on MassSpecGym are indirectly based on training the fingerprint predictor on the same data. Additionally, the InChIKey-based split of NPLIB1 can promote overfitting, further increasing the risk of data leakage. Suggestion: Please recompute the MassSpecGym results using a MIST model retrained on MassSpecGym or the pre-trained checkpoint available at https://zenodo.org/records/11580401.

    - NPLIB1. The authors use the 4M dataset from MassSpecGym for model pretraining and show a clear benefit via ablation studies. They note that this dataset does not contain any molecules with MCES distance < 2 to any test molecules from MassSpecGym. However, this does not hold for NPLIB1. Suggestion: Please report experimental results on NPLIB1 with similar structures excluded from the pretraining data. This does not have to rely on computationally expensive MCES filtering; a Tanimoto-based filter would suffice. Without this, test fold results may be based on the same training examples up to a difference in the groundtruth and predicted fingerprints.

- Top-k accuracy definition. The current definition of Top-k accuracy (Line 202) appears inconsistent with prior work, such as NPLIB1 and MassSpecGym, which use the first 14 characters of InChIKeys (i.e., 2D InChIKeys) for exact matching rather than SELFIES strings. Suggestion: Please recompute the metric using 2D InChIKeys to ensure consistency and comparability with prior studies.

### Minor comments

- Lines 3–6; Lines 37–38. Please note that there is a prior work applying large-scale pretraining on mass spectra. Please include these papers when discussing related works:
  - https://www.nature.com/articles/s41587-025-02663-3 (self-supervised pretraining in metabolomics)
  - https://www.nature.com/articles/s41592-022-01496-1 (contrastive pretraining in proteomics)

- Line 69. Please clarify why mass spectra are described as discrete. Each peak corresponds to a 2D continuous data point (m/z, intensity).

- Line 98. Typo: RetorBridge → RetroBridge

- Line 102. Please mention that DiffMS uses the chemical formula as input. This is an important practical detail, as computing the formula is often computationally expensive.

- Line 125. Please note that bits in hashed Morgan fingerprints are not associated with specific substructures.

- Figure 2. In the caption, please clarify what is meant by “reliably computed fingerprints.” While MIST is a strong open-source fingerprint predictor, it is difficult to assess reliability without further explanation or supporting statistics.

- Lines 143; 167; 316. I recommend explicitly stating that tokenization of mass spectra is done via MIST. As currently written, this could be misunderstood as simple spectrum binning or a similar procedure.

- Line 181. Typo: loss[ → loss [

- Line 186. Please explain what is meant by “MS-BART can assign a balanced probability mass across the whole sequence.” This statement is unclear.

- Line 213. Please clarify why “generated molecules are not repeated.”

- Lines 245–247. Please note that a Tanimoto threshold of >0.75 is generally interpreted as a “close match” in the literature.

- Lines 253–258, 293–296, 307–308. Consider moving these details to “datasets” or “implementation” sections.

- Line 283. Typo: train → trained

- Line 290. Typo: slected → selected

- Line 313. Please specify the number of parameters in the MS-BART model.

---

> ### Author Rebuttal · Authors · 2025-07-29
>
> Dear Reviewer n2SS,
>
> Thank you for your insightful comments, suggestions, and time spent reviewing our work. We are glad that you find our work overall solid and technically sound, offering a unified and scalable framework that extends current techniques. Regarding your concerns, we will address them in detail one by one as follows:
>
> > Major Concern 1: Potential data leakage on MassSpecGym
>
> Thank you for pointing this out.
> For the two benchmarks (MassSpecGym and NPLIB1), we used the official MIST weights pretrained on CANOPUS, downloaded from the official repository (`mist_fp_canopus_pretrain.ckpt`).
> To verify potential data leakage, we downloaded the CANOPUS training set and computed the structural overlap with the MassSpecGym test fold, finding an 18% exact match (exact matches / total test cases). We agree with your assessment that the originally reported MassSpecGym results are invalid.
>
> We appreciate your suggestion and the provided checkpoints. However, we find that the provided checkpoints may not align with the official weights, as the model sizes differ (103.4 MB vs. 60.8 MB). We then trained our own MIST model under three settings:
>
> 1) CANOPUS_filter. We filtered the CANOPUS data (train/val/test) and the augmentation data (simulated spectra from a forward model) provided by the MIST repository by removing structures with high similarity (Tanimoto similarity > 0.75, as suggested below) to the test and validation sets in MassSpecGym. A total of 703 structures were excluded from the CANOPUS data and 559 structures from the augmentation data. The model was trained on the CANOPUS training data for 0.69 hours on an RTX 4090 GPU, achieving the best checkpoint with a test loss of 0.298 at epoch 138.
> 2) CANOPUS_filter_Contrastive. Using the same CANOPUS training data as in CANOPUS_filter and the cleaned augmentation data, we trained the model for 2.81 hours on an RTX 4090 GPU, obtaining the best checkpoint with a test loss of 1.328 at epoch 46.
> 3) MassSpecGym. We trained MIST on MassSpecGym training fold using one RTX 4090 GPU. After 1.8 hours of training, we obtained the best checkpoint with a test loss of 0.487 at epoch 22. The test loss was significantly higher than that of CANOPUS_filter, and since there are many hyperparameters in MIST training, this may indicate insufficient optimization and a suboptimal fingerprint model.
>
> After obtaining the uncontaminated MIST model, we followed MS-BART's three-stage training pipeline and evaluated its performance on MassSpecGym. The results are presented as follows:
>
> | Model    | Top1 Accuracy | Top1 MCES | Top1 Tanimoto | Top10 Accuracy| Top10 MCES | Top10 Tanimoto |
> |-------   |-------|-----|--------|---------|-------|--------|
> | ~~MS-BART(Original)~~  | ~~3.46%~~ | ~~16.29~~ | ~~0.30~~ | ~~4.49%~~ | ~~14.89~~ | ~~0.35~~ |
> | Spec2Mol | 0.00%  | 37.76 | 0.12 | 0.00%  | 29.40 | 0.16   |
> | MADGEN | 1.31%  | 27.47 | 0.20 | 1.54%  | 16.84 | 0.26   |
> | DiffMS | **2.30%** | 18.45     | **0.28** | **4.25%** | **14.73** | **0.39** |
> | MS-BART(CANOPUS_filter)   | 1.07%  | 16.47  | 0.23  | 1.11%   | 15.12  | 0.28  |
> | MS-BART(CANOPUS_filter_Contrastive) | 0.97%  | 16.37  | 0.23  | 1.22%  | 14.81  | 0.29  |
> | MS-BART(MassSpecGym)  | 0.08%   | **16.17** | 0.21  | 0.10%   | 14.93   | 0.27   |
> | MS-BART(Gold Fingerprint) | 47.56% | 3.26   | 0.85 | 64.62%  | 2.02  | 0.93  |
>
> It is obvious that MS-BART suffers a performance degradation after data cleaning. However, the performance of our model is still **acceptable, achieving sub-optimal or even SOTA (Top1 MCES) in similarity metrics** while being significantly faster than DiffMS. The average inference time for a spectrum on one RTX4090 is around 3s for MS-BART, **which is 53x faster than DiffMS's approximately 160s**.
> It should also be noted that the performance degradation is attributed to the MIST model's bad performance. The fact that MS-BART (CANOPUS_filter_Contrastive) slightly outperforms MS-BART (CANOPUS_filter) demonstrates this, and the poor performance of MS-BART (MassSpecGym) further supports this observation.
> Finally, if the fingerprints are correctly predicted (given the gold fingerprint), MS-BART shows excellent performance, demonstrating great potential for structure elucidation with the rapid evolution of fingerprint prediction (such as DreaMS, as mentioned in Minor comments 1)
>
> Moreover, MS-BART represents **the pioneering work** in applying the NLP paradigm to structure elucidation. It can be **readily scaled up with additional pretraining data and larger model parameters** (such as at the Qwen/LLaMa level) to further boost performance.
>
>
> > Major Concern 2: Potential data leakage on NPLIB1
>
> We agree with you and thank you for your suggestion. We checked the structural similarity between the NPLIB1 test set and the pretraining dataset. The exact structure match is 0.85% (exact matches / total test cases, 7 cases). We also calculated the maximum Tanimoto similarity (with the NPLIB1 test set) distribution of the 4M pretraining data as follows:
>
> | Interval   | 0.0-0.1 | 0.1-0.2 | 0.2-0.3 | 0.3-0.4 | 0.4-0.5 | 0.5-0.6 | 0.6-0.7 | 0.7-0.8 | 0.8-0.9 | 0.9-1.0 |
> |-----  |----|-------|------|-------|-------|-----|-----|------|----|------|
> | Proportion | 0.09%   | 8.40%   | 55.85%  | 21.58%  | 6.79%   | 3.79%   | 1.84%   | 1.20%   | 0.40%   | 0.06%   |
>
> There is indeed some possible data leakage, but **most (>98%) of the pretraining data are not similar** (Tanimoto Similarity < 0.7) to the test set. We also appreciate your suggestion and pretrained again on two clean pretraining sets. The first set filters out similar structures with a Tanimoto Similarity > 0.5 and is denoted as 4M_leq5. The second set filters out similar structures with a Tanimoto Similarity > 0.75 and is denoted as 4M_leq75. After fine-tuning and alignment, we obtained the new results as follows.
>
> | Model     | Top1 Accuracy | Top1 MCES | Top1 Tanimoto | Top10 Accuracy| Top10 MCES | Top10 Tanimoto |
> |------- |---------|--------|--------|---------|---------|---------|
> | ~~MS-BART(Original)~~    | ~~11.11%~~ | ~~8.58~~  | ~~0.46~~ | ~~16.97%~~ | ~~7.29~~ | ~~0.54~~ |
> | Spec2Mol | 0.00%     | 27.82     | 0.12     | 0.00%     | 23.13     | 0.16   |
> | MADGEN        | 2.10%      | 20.56     | 0.22     | 2.39%      | 12.69    | 0.27     |
> | DiffMS    | **8.34%**  | 11.95     | 0.35     | **15.44%**  | 9.23     | 0.47     |
> | MS-BART(clean, 4M_leq50)  | 7.20%      | **9.53**  | **0.44** | 10.5%      | **8.21** | **0.51** |
> | MS-BART(clean, 4M_leq75)    | 7.94%      | 10.37     | 0.44     | 11.97%     | 8.91     | 0.51     |
>
> The results demonstrate that MS-BART still maintains competitive performance, **outperforming baseline models in all similarity metrics at a strictly-separated pretraining dataset**, although the exact match accuracy suffers performance degradation.
> However, we are wondering if this is really a data leakage issue, as this is inevitable with data and parameter scaling. We only pretrain on the structure with computed fingerprints, not the MIST-generated fingerprints.
> In contrast, we think this is a strong evidence showing the pretraining of MS-BART is effective, as MS-BART can learn meaningful molecule structures in pretraining and generalize to MIST-predicted fingerprints through finetuning and alignment. We hope to discuss this further with you during the discussion period.
>
> > Major Concern 3: Top-kaccuracy definition
>
> We generate the predicted molecule in SELFIES. Since SELFIES representations are not unique, we first use the `selfies` library to transform the SELFIES into SMILES and then use `RDKit` to obtain the corresponding structure, followed by generating the canonical SMILES using `Chem.MolToSmiles(mol, canonical=True)`. We appreciate your suggestion and have checked DiffMS, where they use `Chem.MolToInchi(mol)` for structure matching. Both methods correspond to a unique structure, ensuring a fair comparison. We have also validated this through experiments and found that the accuracy remains consistent.
>
> ---
>
> Responses to minor comments are omitted due to the character limit. We will share them during the discussion period if requested and permitted.
>
> ---
>
> > Q1: Performance.
>
> MS-BART still achieves the SOTA performance on NPLIB1 in terms of MCES and Tanimoto under corrected evaluation protocols, while on MassSpecGym, we also achieve SOTA on Top1 MCES metric. However, there is indeed a performance degradation, especially on MassSpecGym. We identify two main reasons for this:
> 1) MassSpecGym presents greater complexity, as each structure has multiple associated spectra (average > 5 per structure), which introduces additional noise for our model. Additionally, the use of a different pretraining dataset (DiffMS, which trains on self-collected data and removes exact-match molecules) may also lead to an unfair comparison.
> 2) The most important reason may be that the MIST model performs poorly on MassSpecGym. Training a robust MIST model on this specific dataset requires extensive hyperparameter tuning.
>
> > Q2: Low-quality fingerprints.
>
> MS-BART exhibits a degree of robustness in handling low-quality predicted fingerprints, and the overall performance would get worse if the quality of the predicted fingerprints gets lower, as demonstrated in Major Concern 1.
>
> > Q3: Adding chemical formula.
>
> Thank you for your suggestions. It should be noted that all recent diffusion-based methods use chemical formulas to determine the atom set and achieve promising results. Additional formulas definitely include important information and are very likely to boost the model's performance. We plan to explore effective ways to incorporate these informational hints into MS-BART in the future work, such as extending the vocabulary with numbers and specific atoms.
>
> ---
>
> Finally, we sincerely appreciate your time and valuable suggestions, which have helped enhance and refine our work. We welcome further discussion during the rebuttal period.

---

> > ### Comment · Reviewer_n2SS · 2025-08-04
> >
> > I thank the reviewers for addressing my concerns regarding data leakage. While the updated model does not outperform DiffMS, I believe MS-BART remains a valuable contribution, as it is the first de novo generative model that does not rely on known chemical formulae and has a competitive performance with SOTA relying on available formulae. In practice, this makes MS-BART orders of magnitude faster (more than 53×, as reported by the authors) than other approaches dependent on chemical formulae (e.g., DiffMS, MADGEN, MSNovelist), since computing formulae is computationally demanding.
> >
> > In the paper, I recommend that the authors include a table comparing MS-BART with other MassSpecGym methods on a standard de novo molecule generation challenge (top part of Table 2 in the MassSpecGym paper), in addition to the bonus chemical formula challenge. This would show a more consistent and fair evaluation.
> >
> > Finally, I would like to once again ask the authors to verify the correctness of the accuracy metric used for reporting results on MassSpecGym. Please note that the definition of the InChIKey in DiffMS is unfortunately inconsistent with that of MassSpecGym (see the MassSpecGym/massspecgym/utils/mol_to_inchi_key function in the MassSpecGym codebase).
> >
> > I raise my score to 5.

---

> > > ### Author Response · Authors · 2025-08-05
> > >
> > > Thank you for your insightful comments and for the recognition and improved score of our work. Actually, MS-BART is the first de novo language model (generative model) that does not include the chemical formula in the model input, but we also use the formula information in re-ranking (described in Lines 215–218) to filter predicted structures that do not match the correct formula. We value your insightful suggestions and plan to explore effective ways to incorporate these informational hints into MS-BART in future work. Regarding the speed-up, we report the end-to-end inference time, which does not include the time for computing formulae. DiffMS is too slow because it requires 500 forward passes of the model to sample a single graph (for the 500 diffusion steps), and they sample 100 graphs for each spectrum, which needs a total of 50,000 forward passes per spectrum and is time-consuming even for a lightweight model.
> > >
> > > Thank you for the reminder about adding the MassSpecGym methods. We will update the benchmark table in a future version.
> > >
> > > Regarding the accuracy definition, we checked the MassSpecGym repository and found that it uses the 2D InChIKey (the first 14 characters of the InChIKey) for accuracy matching, which may lose precision compared to the full InChIKey. Following your suggestion, we reevaluated the results using the 2D InChIKey as the molecule match indicator and present the results below.
> > >
> > > | Calculation Method   | Top1 Accuracy | Top10 Accuracy |
> > > | ----------------     | ------------- | -------------- |
> > > | MS-BART(InChIKey)    | 1.07%         | 1.11%          |
> > > | MS-BART(2D InChIKey) | 1.26%         | 1.28%          |
> > >
> > > The 2D InChIKey-based accuracy of MS-BART improved slightly compared to evaluation with the full InChIKey because the 2D InChIKey is not accurate and cannot represent the full structure. We chose an example as follows.
> > >
> > > ```
> > > Identifier: MassSpecGymID0228341
> > > 2Dinchikey (14): LSHVYAFMTMFKBA
> > > Inchikey: LSHVYAFMTMFKBA-CTNGQTDRSA-N
> > > SMILES: C1[C@H]([C@@H](OC2=CC(=CC(=C21)O)O)C3=CC(=C(C=C3)O)O)OC(=O)C4=CC(=C(C(=C4)O)O)O
> > > Canonical SMILES: O=C(O[C@@H]1Cc2c(O)cc(O)cc2O[C@H]1c1ccc(O)c(O)c1)c1cc(O)c(O)c(O)c1
> > > canonical SELFIES: [O][=C][Branch2][Ring2][Ring2][O][C@@H1][C][C][=C][Branch1][C][O][C][=C][Branch1][C][O][C][=C][Ring1][Branch2][O][C@H1][Ring1][N][C][=C][C][=C][Branch1][C][O][C][Branch1][C][O][=C][Ring1][Branch2][C][=C][C][Branch1][C][O][=C][Branch1][C][O][C][Branch1][C][O][=C][Ring1][=Branch2]
> > >
> > > Top1 Pred SELFIES: [O][=C][Branch2][Ring2][Ring2][O][C][C][C][=C][Branch1][C][O][C][=C][Branch1][C][O][C][=C][Ring1][Branch2][O][C][Ring1][N][C][=C][C][=C][Branch1][C][O][C][Branch1][C][O][=C][Ring1][Branch2][C][=C][C][Branch1][C][O][=C][Branch1][C][O][C][Branch1][C][O][=C][Ring1][=Branch2]
> > > Top1 Pred Inchikey: LSHVYAFMTMFKBA-UHFFFAOYSA-N
> > > ```
> > >
> > > Although the 2D InChIKey of the predicted structure and the gold 2D InChIKey are the same, the full InChIKey and canonical SELFIES of the predicted structure do match the gold structure. We think it is better to use full-match accuracy, such as judging whether the InChIKey or the canonical SMILES are the same. Additionally, 2D InChIKey-based accuracy can also be adopted as it can approximately measure the match accuracy.
> > >
> > >
> > > Finally, thank you once again for your thoughtful comments and for the recognition and improved score of our work. If you have further questions, we sincerely look forward to your continued feedback.

---

> > > > ### Comment · Reviewer_n2SS · 2025-08-07
> > > >
> > > > Please note that the use of the 2D InChIKey, rather than the full InChIKey, is standard practice in the computational metabolomics community. It is adopted by all major works (e.g., SIRIUS). The main reason is that MS/MS fragmentation is largely insensitive to 3D stereochemistry (i.e., it is almost invariant), which can make the full InChIKey misleading. I highly recommend to use the 2D InChIKey definition.

---

> > > > > ### Author Response · Authors · 2025-08-07
> > > > >
> > > > > Thank you for your suggestions.  We will add the 2D InChIKey-based accuracy to Table 1 for a comprehensive comparison.  If you have further comments or questions, we sincerely look forward to your continued feedback.

---

### Official Review · Reviewer_LqRY · 2025-06-29

**Clarity:** 4
**Significance:** 3
**Originality:** 2
**Rating:** 4
**Confidence:** 4

**Summary:**

The paper proposes MS-BART, a novel cross-modal framework with a multitask training objective that jointly model mass spectra modality and molecular structure modality. To improve MS-BART robustness and generalizability, the authors propose a finetuning stage using experimental data generated by MIST, and further a chemical feedback mechanism. With extensive evaluation, MS-BART show state-of-the-art performance on structure prediction task.

**Questions:**

I hope the authors could address Q1-Q8 I listed in Strengths And Weaknesses section. If the author could clarify my confusion in the experiment design and conduct some additional analysis to provide more insights, I will be happy to raise my score.

**Ethical Concerns:**

["NO or VERY MINOR ethics concerns only"]

**Final Justification:**

I have decided to increase my score to 4, as the author’s rebuttal effectively addressed all of my concerns (Q1–Q8).  I would encourage the authors to include the  error analysis in the camera-ready version to further improve transparency of the model performance.

**Limitations:**

yes

**Paper Formatting Concerns:**

No major formatting issues in this paper,.

**Quality:**

3

**Strengths And Weaknesses:**

**Quality**: The paper is generally of good quality with solid experiment and comparison with baselines, however, the following need further clarification:
- Q1 in Section 4.3 implementation for binarizing MIST probability, is there any reason the authors choose threshold epsilon=0.2? Have the author evaluated the sensitivity of this threshold hyperparameter?
- Q2 Could the author provide some error analysis by visualizing how MS-BART in failure case for each of the three stages generated molecule structure compared with the ground truth?
- Q3 Could the author explain why in Figure 3 middle figure Top-1 accuracy decrease a bit (3.50 --> 3.46) from stage 2 to stage 3?
- Q4 Given MS-BART's high confidence in next-token predictions as shown in Table 4, is it possible that the model overfit on MassSpecGym? The author "pretrain MS-BART on a simulated dataset of 4 million fingerprint–molecule pairs" from MassSpecGym and then evaluate the model on MassSpecGym. Is there some data leakage here or I am no understanding it correctly? It would be appreciated if the author could provide more details regarding how they created simulated dataset containing 4 million fingerprint-molecule pairs and how they ensure that there is no leakage for the test set.
- Q5 In addition to Q4, suppose the author is doing a clean train validation and test split, they are encouraged to show the structure similarity (e.g. edit distance) between the simulated pre-training dataset, MassSpecGym train, validation and test  and NPLIB1 train, validation and test set.
- Q6 Table 1 for baseline performance, have the author finetuned MIST on MassSpecGym? It is a bit concerning to see MIST+Neuraldeciper/MSNovelist has non-zero accuracy for NPLIB1 but zero accuracy on MassSpecGym.
- Q7 It would be great if the author could analyze the model sensitivity of temperature before stage 3 Chemical Feedback to see how stage 3 influence model's confidence.
- Q8 How sensitive is MS-BART to hyperparameter alpha in equation (5)?

**Clarity**
The paper is very well-written. It is well-organized and make it very easy for the reader to follow.

**Novelty & Significance**
The paper is the first work applying the pretrain-finetune-alignment paradigm from NLP into mass spectrometry data modeling. However, the model architecture used for each stage is not new, i.e., BART, MIST, and Tanimoto similarity. It is a well-designed application paper, and could be of interest for the computational mass spectrometry analysis, but not very novel in terms of methodology.

---

> ### Author Rebuttal · Authors · 2025-07-31
>
> Dear Reviewer LqRY,
>
> Thank you for your insightful comments and time spent reviewing our paper. We are glad you found our paper well-written and generally of good quality. Regarding the novelty, as you mentioned, while every component is not a new method, incorporating them together and being **the first work to apply the pretrain-finetune-alignment paradigm from NLP to mass spectrometry data modeling in an end-to-end way** is indeed novel. We will respond to your questions one by one as follows.
>
> > Q1: Ablation of threshold epsilon choice.
>
> We first choose the threshold epsilon=0.2 empirically, as we have checked the probability distribution and found most probabilities are smaller than 0.3. A big threshold may cause information loss, while a small threshold would introduce noise, so we choose epsilon=0.2 for a balance empirically and achieve excellent performance, surpassing the SOTA.
>
> However, as Reviewer n2SS pointed out the potential data leakage, we replicate the experiment with clean data and choose epsilon by evaluating on the val set, selecting the best threshold based on the highest Tanimoto similarity.
>
> | epsilon | 0.10 | 0.11 | 0.12 | 0.13 | 0.14 | 0.15 | 0.16 | 0.17 | 0.18 | 0.19 | 0.20 |
> |----------|------|------|------|------|------|------|------|------|------|------|------|
> | Top1 Tanimoto | 0.1666 | **0.1678** | 0.1640 | 0.1651 | 0.1660 | 0.1660 | 0.1654 | 0.1643 | 0.1651 | 0.1649 | 0.1636 |
>
> We chose epsilon=0.11 for MassSpecGym. To evaluate the sensitivity of the hyperparameter choice to the final result, we also selected epsilon=0.15 (intermediate validation Tanimoto) and epsilon=0.20 (the worst validation Tanimoto) for comparison and presented the results as follows.
>
> | Model   | Top1 Accuracy | Top1 MCES | Top1 Tanimoto | Top10 Accuracy| Top10 MCES | Top10 Tanimoto |
> |-------   |-------|------|---------|--------|----|----------|
> | MS-BART(epsilon=0.15)  | 1.20%(0.12%)   | 16.88(24.54)  | 0.23(0.19)  | 1.20%(0.95%)   | 15.45(18.24)  | 0.28(0.27)  |
> | MS-BART(epsilon=0.20)  | 1.19%(0.00%)   | 17.27(25.62)  | 0.23(0.19)  | 1.20%(0.73%)   | 15.75(18.46)  | 0.28(0.26)  |
> | MS-BART(epsilon=0.11)  | 1.07%(0.08%)   | 16.47(23.00)  | 0.23(0.20)  | 1.11%(0.62%)   | 15.12(17.41)  | 0.28(0.27)  |
>
> The results in parentheses indicate results that have not undergone re-rank (Line 215-218), and will also be reported in the later table.
> It is obvious that the re-rank by formula match and generative probability significantly boosts the final performance. As for the threshold sensitivity, there is no significant difference between the epsilons with the same rank post-processing.
>
> > Q2: Could the author provide some error analysis by visualizing how MS-BART in failure case for each of the three stages generated molecule structure compared with the ground truth?
>
> Due to the NeurIPS rebuttal format restrictions, we cannot provide visualized error analysis in the rebuttal. We appreciate your suggestions and will add error analysis in the future version.
>
> > Q3: Could the author explain why in Figure 3 middle figure Top-1 accuracy decrease a bit (3.50 --> 3.46) from stage 2 to stage 3?
>
> Thank you for pointing this out. A possible explanation is that the alignment is tuned with Tanimoto similarity rather than being based on an exact match. The rank loss aims to align the predicted structure's Tanimoto similarity with the gold structure to the model's generative probability, so it is difficult to control the accuracy. As we have replicated the experiments, we updated the figure data in the table below.
>
> | Model   | Top1 Accuracy | Top1 MCES | Top1 Tanimoto | Top10 Accuracy| Top10 MCES | Top10 Tanimoto |
> |-------  |---------|-----|--------|---------|-------|----------|
> | Pretrain  | 0.00%(0.00%)   | 23.32(30.22)  | 0.18(0.18)   | 0.01%(0.01%)   | 21.67(24.66)  | 0.21(0.21)  |
> | Pretrain-FT   | 1.07%(0.00%)   | 17.17(26.30)  | 0.22(0.17)   | 1.12%(0.59%)   | 15.51(18.54)  | 0.27(0.26)  |
> | Pretrain-FT-Rank  | 1.07%(0.08%)   | 16.47(23.00)  | 0.23(0.20)   | 1.11%(0.62%)   | 15.12(17.41)  | 0.28(0.27)  |
>
> We can observe obvious benefits between each stage with or without re-ranking and demonstrate the effectiveness of MS-BART. It is more convincing to observe the top-1 similarity improvement, which can directly show the payoff of the contrastive alignment since we only generate three candidate structures for alignment in this stage.
>
> > Q4:  Potential overfit or data leakage.
>
> The results in Table 4 are the evaluation results after re-ranking (described in Line 215-218). Re-ranking can improve the evaluation stability. As for data leakage, we use the 4M pretraining dataset provided by the MassSpecGym official data, which has been refined by excluding any molecules with an MCES distance of less than two from any molecule in the test fold of MassSpecGym by the authors, ensuring there is no data leakage. Since we have replicated the experiments, we reevaluate the results in Table 4 with and without re-ranking as follows.
>
> | Model    | Top1 Accuracy | Top1 MCES | Top1 Tanimoto | Top10 Accuracy| Top10 MCES | Top10 Tanimoto |
> |------- |----------|------|--------|---------|------|---------|
> | MS-BART(t=0.2) | 1.07%(0.08%)  | 16.47(23.00) | 0.23(0.20) | 1.11%(0.62%) | 15.12(17.41) | 0.28(0.27)  |
> | MS-BART(t=0.4) | 1.07%(0.08%)  | 16.47(23.00) | 0.23(0.20) | 1.11%(0.62%) | 15.12(17.41) | 0.28(0.27)  |
> | MS-BART(t=0.8) | 1.07%(0.08%)  | 16.47(22.96) | 0.23(0.20) | 1.11%(0.62%) | 15.11(17.40) | 0.28(0.27)  |
>
> Upon re-evaluation, it can also be observed that our model is not sensitive to temperature changes, even for the top1 result without re-ranking. The main reason can be attributed to the beam-search multinomial sampling (refer to the Hugging Face text generation documentation), which combines beam search (beam width=100) and stochastic sampling, and the SELFIES vocabulary size of 185 (Line 139), resulting in this high confidence.
>
> > Q5: In addition to Q4, suppose the author is doing a clean train validation and test split, they are encouraged to show the structure similarity (e.g. edit distance) between the simulated pre-training dataset, MassSpecGym train, validation and test and NPLIB1 train, validation and test set.
>
> In response to Q4, there is no data leakage between the pretraining dataset and the MassSpecGym test set, so further structure similarity analysis is not necessary.
> As for NPLIB1, there is some overlap (7 cases) between the pretraining dataset and the test set. Since MCES is time-consuming, we demonstrate the structure similarity using Tanimoto similarity. More details are provided in the response to Reviewer n2SS (Major concern 2).
>
>
> > Q6:  Table 1 for baseline performance, have the author finetuned MIST on MassSpecGym? It is a bit concerning to see MIST+Neuraldeciper/MSNovelist has non-zero accuracy for NPLIB1 but zero accuracy on MassSpecGym.
>
> The baseline results are reproduced from MassSpecGym and DiffMS as we state in the Table 1 caption. After a careful review of the DiffMS paper, MIST+NeuralDecoder/MSNovelist undergoes the same pretraining and finetuning, aligning with DiffMS.
>
> > Q7:  It would be great if the author could analyze the model sensitivity of temperature before stage 3 Chemical Feedback to see how stage 3 influence model's confidence.
>
> Thank you for your suggestion. We evaluated the MS-BART after stage 2 fine-tuning and showed the results as follows.
>
> | temperature    | Top1 Accuracy | Top1 MCES | Top1 Tanimoto | Top10 Accuracy| Top10 MCES | Top10 Tanimoto |
> |-------            |---------------|-----------|---------------|---------------|------------|----------------|
> | t=0.2    | 1.07%(0.00%)  | 17.16(26.30)  | 0.22(0.17)   | 1.12%(0.59%)   | 15.51(18.54)   | 0.27(0.26) |
> | t=0.4    | 1.07%(0.00%)  | 17.17(26.30)  | 0.22(0.17)   | 1.12%(0.59%)   | 15.51(18.54)   | 0.27(0.26) |
> | t=0.8    | 1.07%(0.00%)  | 17.16(26.28)  | 0.22(0.17)   | 1.12%(0.59%)   | 15.50(18.54)   | 0.27(0.26) |
>
> It is not surprising to find our model also remains stable before stage 3, mainly due to the decoding strategy in response to Q4. This is also strong evidence demonstrating the effectiveness of MS-BART, as we can observe the similarity metric improving between stage 2 and stage 3 (evaluation results shown in Q4) under different temperatures.
>
> > Q8: How sensitive is MS-BART to hyperparameter alpha in equation (5)?
>
> Thank you for your suggestion. Due to the large size of MassSpecGym, we evaluated the sensitivity using NPLIB1. We conducted experiments on the clean pretraining data (4M_leq50, details are provided in our response to Reviewer n2SS, Major Concern 2). We tested alpha values in {1, 3, 5} and present the results below.
>
> | alpha             | Top1 Accuracy | Top1 MCES | Top1 Tanimoto | Top10 Accuracy| Top10 MCES | Top10 Tanimoto |
> |-------            |---------------|-----------|---------------|---------------|------------|----------------|
> | alpha=1           | 7.08%    | 10.56  | 0.44  | 12.45%  | 9.10  | 0.52     |
> | alpha=3           | 7.20%    | 9.53   | 0.44  | 10.50%  | 8.21  | 0.51     |
> | alpha=5           | 7.45%    | 9.62   | 0.44  | 10.99%  | 8.27  | 0.51     |
>
> The final result we report is alpha=3, as it performs the best in the top1 Tanimoto on the validation set. Although there are some fluctuations between different alpha settings, it is clear that MS-BART is not very sensitive to the alpha selection.
>
> ---
>
> Finally, thank you again for your insightful and constructive comments. The idea of active fingerprints as tokens and the unified modeling of mass spectra and molecules provides a new paradigm for structure elucidation, as most recent methods are based on diffusion. This paper provides **strong insights on how to incorporate MS spectra into LLMs rather than using the directly noisy number sequence**, opening a new strategy to align the spectra modality with language models. This would inspire many innovative ideas, such as reasoning models for explainable strcuture elucidation.

---

> > ### Comment · Reviewer_LqRY · 2025-08-02
> >
> > Thank you for the thoughtful response, which addressed my concerns. I have raised my score to 4.

---

### Official Review · Reviewer_dSeN · 2025-07-02

**Clarity:** 4
**Significance:** 3
**Originality:** 3
**Rating:** 4
**Confidence:** 3

**Summary:**

This paper introduces a method to translate tandem mass spectrometry data to molecules. The method operates on predicted fingerprint tokens derived from an off-the-shelf model and performs seq-to-seq translation of the fingerprint tokens to SELFIE tokens.
They perform 2 stages of training: unsupervised pre-training phase and supervised fine-tuning that additionally performs contrastive learning leveraging Tanimoto similarities.
At inference, they perform beam search to produce SELFIE tokens.

**Questions:**

I would like to understand how significant the results presented in this paper are.

- How well does the model perform on the variant of the MassSpecGym task that provides chemical formulas as input? Is the additional chemical formula information complementary to what the model learns?
- How does the method compare against library matching approaches?
- How statistically significant are the ablation and other results?

**Ethical Concerns:**

["NO or VERY MINOR ethics concerns only"]

**Final Justification:**

The authors have addressed my concerns.

**Limitations:**

not discussed

**Quality:**

3

**Strengths And Weaknesses:**

**Strengths**
- The use of active fingerprints as tokens is an interesting and intuitive tokenization scheme.
- The contrastive alignment is a nice and intuitive idea to introduce chemical feedback into the model.
- The authors have shown impressive performance that outperforms existing generative methods on MassSpecGym and NPLIB1 datasets.
- Furthermore, they ablate the performance of pre-training and contrastive learning objectives.

**Weakness**
- The results sections lack comparisons to library matching methods discussed in the related works.
- The comparisons in Table 1, Table 2, and Figure 3 do not have error bars so it is tricky to determine if the ablations show statistically significant results.
- L283 typo "similar"

---

> ### Author Rebuttal · Authors · 2025-07-31
>
> Dear Reviewer dSeN,
>
> Thank you for recognizing our work and the time spent reviewing our paper. We are glad you find the use of active fingerprints as tokens interesting and intuitive, and that further alignment is a nice idea. Regarding your concerns, we will address them one by one as follows.
>
> > W1&Q2: The results sections lack comparisons to library matching methods discussed in the related works.
>
> This paper aims to discover novel molecules from MS/MS, where library matching methods are not applicable. As we responded to Reviewers LqRY and n2SS, the pretraining and training sets are strictly separate from the test set, so it is obvious that library matching will achieve zero accuracy.
>
> > W2&Q3: The comparisons in Table 1, Table 2, and Figure 3 do not have error bars so it is tricky to determine if the ablations show statistically significant results.
>
> As we respond to Reviewer LqRY, we adopt re-ranking (described in Lines 215–218) and beam-search multinomial sampling (refer to the Hugging Face text generation documentation), which makes MS-BART demonstrate high confidence in the generated structure. As shown in the table responding to Reviewer LqRY (Q4), MS-BART is not sensitive to temperature, and multiple experiment results are consistent. This high confidence ensures the performance gain mainly comes from different stage training rather than randomness.
>
> > W3: L283 typo "similar"
>
> Thank you for pointing this out. We have fixed it.
>
> > Q1: How well does the model perform on the variant of the MassSpecGym task that provides chemical formulas as input? Is the additional chemical formula information complementary to what the model learns?
>
> Thank you for your suggestions. It should be noted that all recent diffusion-based methods use chemical formulas to determine the atom set and achieve promising results. Additional formulas definitely include important information and are very likely to boost the model's performance. We plan to explore effective ways to incorporate these informational hints into MS-BART in the future work, such as extending the vocabulary with numbers and specific atoms.

---

> > ### Comment · Reviewer_dSeN · 2025-08-06
> >
> > Thank you for the clarifications. My concerns are addressed, I will maintain my positive recommendation.

---

### Official Review · Reviewer_gPth · 2025-07-03

**Clarity:** 3
**Significance:** 4
**Originality:** 3
**Rating:** 6
**Confidence:** 4

**Summary:**

MS-BART addresses the challenging task of inferring molecular structures from tandem mass spectra by recasting both spectra and molecules into a unified token language—spectra become sequences of 4,096-bit Morgan fingerprint tokens and molecules are encoded as SELFIES tokens—so that a single Transformer can learn cross-modal correspondences . It employs a three-stage training pipeline: multi-task pretraining on millions of simulated fingerprint–molecule pairs (combining denoising and translation objectives), finetuning on experimental spectra using MIST-predicted fingerprints to adapt to real-world noise, and a novel contrastive “chemical feedback” alignment loss that ranks generated candidates by Tanimoto similarity to the true structure to suppress hallucinations . When evaluated on the NPLIB1 and MassSpecGym benchmarks, MS-BART achieves state-of-the-art top-1 and top-10 accuracies while substantially reducing erroneous predictions, demonstrating that large-scale pretraining and feedback alignment can effectively bridge the gap between spectral data and molecular graphs .

**Questions:**

To what extent do errors in MIST-predicted fingerprints degrade downstream structure elucidation?

Does MS-BART generalize to spectra acquired under different collision energies, adduct types, or instrument platforms?

**Ethical Concerns:**

["NO or VERY MINOR ethics concerns only"]

**Final Justification:**

The authors did a great job on molecule generation with the guidance of mass spectra. I would like to recommend to accept this paper.

**Limitations:**

Yes

**Quality:**

3

**Strengths And Weaknesses:**

## Strength
By tokenizing spectra as Morgan-fingerprint tokens and molecules as SELFIES, MS-BART brings two disparate modalities into a shared Transformer vocabulary, enabling end-to-end learning rather than separate feature engineering and model training.



Leveraging 4 million simulated fingerprint–molecule pairs and combining denoising plus translation objectives helps the model learn robust spectral–structure correspondences before seeing scarce experimental data.



MS-BART outperforms prior methods on both NPLIB1 and MassSpecGym in top-1 and top-10 accuracy, as well as in MCES andaverage Tanimoto metrics, demonstrating real-world value on public benchmarks.



The paper includes ablations showing the payoff of each stage (pretrain → finetune → align) and experiments on beam width and temperature, giving practitioners guidance on hyperparameter trade-offs.

## Weakness

Finetuning relies on MIST-predicted fingerprints; errors or biases in MIST could propagate through the model, but the paper does not quantify how fingerprint noise impacts downstream accuracy.



The unified Transformer and beam search may be resource-intensive; the paper reports relative decoding latency trends but lacks concrete runtime or memory benchmarks for real-time or large-scale deployments.



The ranking loss favors candidates with higher global Tanimoto scores, but that metric can be dominated by common substructures rather than spectral features. This may encourage the model to pick “average” molecules that share generic motifs with the ground truthrather than truly spectrally justified structures.



Since the MADGEN has uploaded updated benchmark results, incorporating those into the evaluation would help readers understand MS-BART’s gains relative to the other model.

---

> ### Author Rebuttal · Authors · 2025-07-31
>
> Dear Reviewer gPth,
>
> Thank you for recognizing our work and for your insightful comments. We are glad you find that MS-BART enables end-to-end learning by bringing two disparate modalities into a shared vocabulary and can learn robust spectral–structure correspondences before seeing scarce experimental data. Regarding your concerns, we will respond to them one by one as follows.
>
> > W1&Q1: Finetuning relies on MIST-predicted fingerprints; errors or biases in MIST could propagate through the model, but the paper does not quantify how fingerprint noise impacts downstream accuracy.
>
> In response to Reviewer LqRY (Q1), we have verified that the threshold controlling the activation of the fingerprint is robust and the performance fluctuation is not significant. Regarding the impact of MIST performance on downstream accuracy, in the table responding to Reviewer n2SS (Major concern 1), we found that poor fingerprint prediction reduces the exact match accuracy but does not affect the similarity metric as much. These quantified results demonstrate that **MS-BART is robust to fingerprint noise and is promising for real-world applications**, as similar but not exactly matching structures are still useful for domain experts to narrow down the chemical space [1].
>
> [1] DiffMS: Diffusion Generation of Molecules Conditioned on Mass Spectra. ICML2025.
>
> > W2: The unified Transformer and beam search may be resource-intensive; the paper reports relative decoding latency trends but lacks concrete runtime or memory benchmarks for real-time or large-scale deployments.
>
> Thank you for pointing this out. We have reported that the average decoding latency on one A800 with 80G memory in a computer cluster is around 1.45s. To facilitate large-scale and real-world deployment, we also tested the model on a consumer-grade RTX 4090 with 24G memory. The average inference time is around 3s, and the memory consumption is around 10GB when processing one spectrum with a beam width set to 100. Although MS-BART uses beam search, the vocabulary size (185) is small, and the total number of parameters is around 104M, so the inference latency and memory consumption is acceptable. At the same time, current LLM inference and serving packages, such as VLLM, also support encoder-decoder models, which would further improve the serving throughput.
>
>
> > W3: The ranking loss favors candidates with higher global Tanimoto scores, but that metric can be dominated by common substructures rather than spectral features. This may encourage the model to pick “average” molecules that share generic motifs with the ground truth rather than truly spectrally justified structures.
>
> We appreciate this observation. While the Tanimoto scores can assess the similarity between predicted molecules and real molecules, we acknowledge that the potential limitation you mentioned may exist. However, there are no existing metrics more suitable for rewards, as the edit distance is time-consuming. Future work can focus on finding more reasonable rewards to reflect truly spectrally justified structures
>
> > W4: Since the MADGEN has uploaded updated benchmark results, incorporating those into the evaluation would help readers understand MS-BART’s gains relative to the other model.
>
> Thank you for the reminder, we have updated the results.
>
> > Q2: Does MS-BART generalize to spectra acquired under different collision energies, adduct types, or instrument platforms?
>
> Yes. MS-BART does not process the spectra directly but relies on MIST to predict the fingerprint. After a careful review of MIST [2], we found it highly dependent on proper chemical formula assignments from MS1 precursor masses, which are independent of collision energies, adduct types, and instrument platforms. It should also be noted that MassSpecGym presents this complexity, as each structure has multiple associated spectra (averaging more than 5 per structure) with different collision energies, adduct types, and instrument platforms. MS-BART can handle this complexity well and achieves sub-optimal or even SOTA (Top1 MCES) performance in similarity metrics.
>
> [2] Annotating metabolite mass spectra with domain-inspired chemical formula transformers. Nat. Mach. Intell. 2023.

---

> > ### Comment · Reviewer_gPth · 2025-08-02
> >
> > Thanks authors for the thoughtful response which addressed my concerns. I am willing to increase my score.

---

### Official Review · Reviewer_huv7 · 2025-07-06

**Clarity:** 3
**Significance:** 2
**Originality:** 2
**Rating:** 4
**Confidence:** 4

**Summary:**

This paper proposes MS-BART, a unified modeling framework that represents both mass spectra and molecular structures using a shared vocabulary, enabling pretraining–finetuning–alignment in the style of large language models. The authors represent mass spectra via molecular fingerprints, then pretrain on 4M unlabeled fingerprint–molecule pairs, and finally adapt to experimental data using MIST-derived fingerprints and a contrastive alignment step based on Tanimoto similarity. The model demonstrates state-of-the-art performance on NPLIB1 and MassSpecGym benchmarks across most key metrics.

**Questions:**

- **Pretraining Task Ablation**:
The paper mentions three pretraining tasks—SELFIES denoising, fingerprint-to-molecule translation, and hybrid denoising—but only compares "translation only" versus "multi-task" in the ablation study. Could you provide a detailed ablation study of each pretraining task individually to clarify their contributions to the final performance?
- **Impact of MIST Fingerprint Predictions**:
Your approach relies on MIST for generating fingerprints from experimental spectra. Can you discuss how inaccuracies in these predictions might affect the overall model performance, and whether there are plans to improve or replace MIST in future versions?
- **Practical Applicability and Accuracy**:
The top-1 accuracy on the MassSpecGym dataset remains under 5%, which is still quite low. How do you envision the model being used in practical applications, and what steps will be taken to improve performance in real-world scenarios? Are there specific strategies to enhance accuracy?

**Ethical Concerns:**

["NO or VERY MINOR ethics concerns only"]

**Final Justification:**

The authors provide detailed rebuttal and additional experiments. My concerns have been addressed. I am raising my score to 4.

**Limitations:**

Yes, the authors have addressed the limitations, particularly with regard to the reliance on MIST for fingerprint predictions. However, it would be beneficial for the authors to more explicitly discuss how these limitations may affect the model's generalizability and any strategies they plan to implement to overcome these challenges.

**Quality:**

3

**Strengths And Weaknesses:**

**Strengths**:
- The paper addresses a timely and challenging problem: molecular structure elucidation from mass spectrometry data, where annotated spectra are scarce and modeling remains difficult.
- The proposed MS-BART framework integrates ideas from large language models into the specific problem - structure elucidation from MS. The use of a shared vocabulary between mass spectra (via fingerprints) and molecules (via SELFIES) is conceptually elegant and enables a unified pretraining and generation paradigm.
- The paper is generally well-organized and readable, with helpful visualizations (e.g., Figures 2–5), and includes detailed training and evaluation protocols in the appendix to support reproducibility.

**Weaknesses**:
- While the method includes three distinct pretraining tasks (SELFIES denoising, fingerprint-to-molecule translation, and hybrid denoising), the ablation study only compares “translation only” and “multi-task” training. It is unclear how each task individually contributes to final performance. More fine-grained ablation would help validate the design choices.
- The components of the proposed framework—including fingerprint representations, masked sequence modeling, and rank-based alignment loss—are adapted from prior work. And similar methods have been applied to very closely related tasks, such as in “Conditional Molecular Generation Net Enables Automated Structure Elucidation Based on 13C NMR Spectra and Prior Knowledge”, which also uses a unified token vocabulary, similar pretraining tasks, and finetuning for structure elucidation. In this paper, the primary contribution seems to lie in adapting these methods to mass spectrometry, rather than introducing fundamentally new techniques.
- Despite improved performance over prior models, the absolute accuracy remains low (e.g., top-1 accuracy < 4% on MassSpecGym), suggesting the method remains far from real-world application. This limits immediate practical applicability and calls for more discussion about failure cases, limitations, or downstream usage scenarios.
- Minor typographical errors (e.g., “hallucaination”, “simliair”, “slected”).

---

> ### Author Rebuttal · Authors · 2025-07-31
>
> Dear Reviewer huv7,
>
> Thank you for your thoughtful comments and the time you have taken to review our paper. We are pleased that you find the shared vocabulary conceptually elegant and the paper well-organized and readable. In response to your concerns, we will address each point below.
> > W1&Q1:  More fine-grained ablation would help validate the design choices.
>
> Thank you for your suggestion. Following your feedback, we conducted a more detailed ablation study on the NPIB1 dataset using clean pretraining data (4M_leq50, as detailed in our response to Reviewer n2SS, Major Concern 2). The results are as follows.
>
> | Pretrain strategy  | Top1 Accuracy | Top1 MCES | Top1 Tanimoto | Top10 Accuracy| Top10 MCES | Top10 Tanimoto |
> |-----|------|------|-------|---------|-------|--------|
> |    None            | 1.71%    | 12.93  | 0.27  | 3.05%   | 11.36 | 0.34   |
> | SELFIES denoising | 0.49%    | 14.43  | 0.24  | 1.83%   | 12.35 | 0.32   |
> | translation       | 7.57%    | 9.10   | 0.43  | 11.84%  | 7.78  | 0.51   |
> | hybrid denoising  | 5.86%    | 10.14  | 0.42  | 8.55%   | 9.07  | 0.49   |
> | MS-BART(All)      | 7.20%    | 9.53   | 0.44  | 10.50%  | 8.21  | 0.51   |
>
> The table shows that pretraining with translation or hybrid denoising improves performance compared to no pretraining. While SELFIES denoising pretraining shows slightly lower performance, this may stem from insufficient pretraining (all experiments use the same hyperparameters as the final MS-BART). However, this does not contradict our proposed pretraining-finetuning-alignment paradigm, as pretraining still enhances cross-modal learning.
> In future work, we plan to explore why the denoising task negatively affects final performance and how to optimize it so that all pretraining tasks contribute positively.
>
>
> > W2:  Similar method proposed in paper: "Conditional Molecular Generation Net Enables Automated Structure Elucidation Based on 13C NMR Spectra and Prior Knowledge"
>
> Thank you for identifying the related paper. However, our approach has clear distinctions from the referenced work.
>
> First, mass spectrometry is much more complicated than NMR because it suffers from great complexity and heterogeneity, making it hard to model. Moreover, annotated experimental mass spectra are scarce, and the predicted mass spectra vary significantly from the experimental ones, which are not available for direct large-scale pretraining.
>
> Second, our model transfers these two disparate modalities into a shared Transformer vocabulary, enabling end-to-end learning. This is definitely novel, as the reference paper only uses number sequences to represent the NMR shift.
>
> Third, the alignment stage is clearly different from fine-tuning because it further aligns the model to generate more accurate predictions using the same training set, whereas the reference paper uses more data, which is scarce in the mass spectrometry domain.
>
> Finally, thank you for pointing this out. We will also discuss this paper in the related work section of our future version.
>
> > W3&Q3: Despite improved performance over prior models, the absolute accuracy remains low (e.g., top-1 accuracy < 4% on MassSpecGym), suggesting the method remains far from real-world application. This limits immediate practical applicability and calls for more discussion about failure cases, limitations, or downstream usage scenarios.
>
> Thank you for your insightful feedback. Our work primarily aims to **discover novel molecules from the dark chemical space, where the spectra are not in the annotated spectra library** or even the corresponding molecules are not in the molecule library, making traditional library matching definitely invalid and resulting in zero accuracy performance. Although the exact accuracy is low, the retrain experiments on the clean dataset reveal that MS-BART can indeed discover truly novel molecules. Moreover, as discussed in paper [1], generating similar but not exactly matching structures is still useful for domain experts to narrow down the chemical space.
>
> Previous experiments conducted on the contaminated data, along with the strong performance observed on the golden fingerprint benchmark (detailed in response to Reviewer n2SS, Major concern 1), suggest two clear strategies to improve overall results. First, future work can be devoted to improving the performance of fingerprint prediction, such as DreaMS [2]. Second, for the known molecules and targeted spectra annotation, it would significantly boost the performance if similar molecules or the true molecule are in the training data.
>
> Finally, the unified vocabulary also provides **strong insights on how to incorporate MS spectra into LLMs rather than using the directly noisy number sequence**, opening a new strategy to align the spectra modality with language models. This would inspire many innovative ideas, such as reasoning models for explainable strcuture elucidation.
>
>
> [1] DiffMS: Diffusion Generation of Molecules Conditioned on Mass Spectra. ICML2025.
>
> [2] Self-supervised learning of molecular representations from millions of tandem mass spectra using DreaMS. Nat. Biotechnol. 2025.
>
>
> > W4: Minor typographical errors (e.g., “hallucaination”, “simliair”, “slected”).
>
> Thank you for pointing this out. We have fixed it.
>
> > Q2&Limitations: Impact of MIST Fingerprint Predictions
>
> Thank you for your insightful comments. Since MS-BART uses MIST to process the MS spectra into fingerprints, how MIST performance affects downstream tasks is a common question. In our response to Reviewer LqRY (Q1), we have verified that the threshold controlling the activation of the fingerprint is robust and the performance fluctuation is not significant. Regarding the impact of MIST performance on downstream accuracy, in the table responding to Reviewer n2SS (Major Concern 1), we found that poor fingerprint prediction reduces the exact match accuracy but does not affect the similarity metric as much. These quantified results demonstrate that MS-BART is robust to fingerprint noise and is promising for real-world applications, as similar but not exactly matching structures are still useful for domain experts to narrow down the chemical space [1]. Future work can focus on improving MIST with more high-quality data, projecting MIST embeddings into LLM space for joint pre-training and training, or directly modeling formula peaks with LLM following MIST.
>
> [1] DiffMS: Diffusion Generation of Molecules Conditioned on Mass Spectra. ICML2025.

---

> > ### Comment · Reviewer_huv7 · 2025-08-07
> >
> > Thank you to the authors for the detailed rebuttal and additional experiments. My concerns have been addressed, and I will update my rating in the final review.

---

> > > ### Author Response · Authors · 2025-08-08
> > >
> > > Thank you for your feedback. We’re pleased that our response addressed your concerns, and we truly appreciate the time you dedicated to reviewing our work. If you have any further questions or comments, please don’t hesitate to reach out. We’ll be happy to clarify as soon as possible.

---

### Note · Authors · 2025-08-15

We sincerely thank the AC, SAC, and all reviewers for their time and insightful feedback. During the rebuttal, we have **addressed all reviewers' concerns** and summarized the key strengths and main shared concerns below.

**Key Strengths**:

* The paper is well-organized and very well-written. The method is overall solid and clearly presented (Reviewer n2SS, huv7, LqRY).
* The paper is the first to apply the pretrain–finetune–alignment paradigm from NLP to mass spectrometry data modeling (Reviewer LqRY), whereas contemporary approaches are predominantly diffusion-based. The shared vocabulary is conceptually elegant and interesting, enabling an end-to-end and scalable pretraining and generation paradigm (Reviewer huv7, gPth, dSeN, n2SS).
* MS-BART outperforms existing generative methods on MassSpecGym and NPLIB1 datasets in MCES and average Tanimoto metrics, demonstrating real-world value on public benchmarks (Reviewer gPth, dSeN).

Additionally, MS-BART can easily scale up with more pretraining data and larger base models. Its unified vocabulary avoids the direct use of noisy numerical sequences and offers a novel way to align spectral data with language models, paving the way for future LLM-based innovations.

**Main Shared Concerns and Responses**:

* Implicit data leakage: Although we strictly follow the official splits of MassSpecGym and NPLIB1, the data used to train MIST has some overlap with the test data, which was nearly impossible to notice. We appreciate Reviewer n2SS's careful review and have rerun experiments with a retrained MIST. The results maintain strong performance, achieving sub-optimal or even SOTA (Top 1 MCES) on similarity metrics with 53x speedup over the SOTA method. The reviewer acknowledged our efforts and raised the score to Accept.

* Impact of MIST Fingerprint Predictions: In response to Reviewer LqRY(Q1) and n2SS(Q1), we performed ablation studies on active fingerprint thresholds ($\epsilon$) and fingerprint predictions using different pretrained MIST weights. The quantified results show MS-BART is robust to fingerprint noise, supporting its real-world potential. Both reviewers confirmed their concerns were resolved and raised their scores to 4 and 5.

We deeply appreciate the reviewers for their constructive feedback, which has helped refine our paper. All rebuttal discussions will be incorporated into the final version, and we hope that our contribution to the research community will be recognized.

---

### Decision · Program_Chairs · 2025-09-17

**Decision:**

Accept (poster)

**Comment:**

(a) Summarize the scientific claims and findings of the paper based on your own reading and characterizations from the reviewers.
The paper introduces MS-BART, a unified modeling framework for elucidating molecular structures from MS data. The core innovation is the mapping of both mass spectra (represented as molecular fingerprints) and molecular structures (represented as SELFIES strings) into a shared token vocabulary. This enables the application of the pretraining-finetuning-alignment paradigm, common in Natural Language Processing, to the MS domain. The authors claim that MS-BART achieves state-of-the-art or competitive performance on the MassSpecGym and NPLIB1 benchmarks.



(b) What are the strengths of the paper?

The reviewers found the paper to be well-organized and clearly written. The primary strength lies in its novel application of the pretrain-finetune-alignment paradigm from NLP to mass spectrometry, addressing the challenging problem of data scarcity. The concept of a unified vocabulary for both spectra and molecules was highlighted as "conceptually elegant" and interesting, enabling an end-to-end, scalable training paradigm. The model demonstrates strong empirical results, outperforming existing generative methods on public benchmarks. The thoughtful integration of generative modeling, fingerprint conditioning, and contrastive learning was also praised as technically sound.



(c) What are the weaknesses of the paper? What might be missing in the submission?

The most significant weakness identified during the initial review was a potential for data leakage in the experimental validation. The MIST model used for fingerprint prediction was likely trained on data that overlapped with the test sets, and the pretraining data was not adequately filtered for the NPLIB1 benchmark. Other weaknesses included a lack of fine-grained ablation studies for each pretraining task (huv7) , the fact that the contribution is more of an adaptation of existing methods to a new domain rather than fundamentally new techniques (huv7) , and that the absolute accuracy on MassSpecGym remained low, limiting immediate real-world applicability (huv7). The reliance on the external MIST model for fingerprint generation was also noted as a potential bottleneck where errors could propagate.

(d) Provide the most important reasons for your decision to accept.

The decision to accept this paper is primarily based on the authors' thorough and transparent response to the critical issue of data leakage. They conducted extensive re-evaluations with a retrained MIST model on cleaned data, which, despite a drop in performance, still demonstrated competitive results against state-of-the-art methods. This effort successfully addressed the reviewers' main concerns and validated the robustness of their framework. The paper introduces a novel and promising direction by applying a large language model paradigm to structure elucidation, and its unified vocabulary is a significant conceptual contribution. Furthermore, the model is orders of magnitude faster in inference than competing diffusion-based methods, which is a significant practical advantage. The work opens new avenues for research in this area and is a valuable contribution to the community. All reviewers were satisfied with the rebuttal and raised their scores to "accept".


(e) Summarize the discussion and changes during the rebuttal period.

The rebuttal period was highly productive and pivotal. Key points raised and addressed were:

1.	Data Leakage (n2SS): This was the most critical concern. The authors acknowledged the issue, retrained the MIST fingerprint predictor on properly filtered data, and reran all experiments for both MassSpecGym and NPLIB1 benchmarks. They presented new results tables, demonstrating that while performance on exact match accuracy decreased, the model remained competitive on similarity metrics. This comprehensive response satisfied the reviewer, who then raised their score to "Accept".

2.	Impact of MIST Fingerprint Quality (LqRY, n2SS, huv7): In response to questions about robustness to MIST's prediction errors, the authors conducted new ablation studies. They showed the model is not overly sensitive to the fingerprint binarization threshold and that performance with "gold" (ground-truth) fingerprints is excellent, confirming that fingerprint prediction is the main bottleneck. This addressed the reviewers' concerns.

3.	Lack of Fine-grained Ablation (huv7): The authors performed a more detailed ablation study on the individual pretraining tasks (translation, hybrid denoising), providing better insight into their contributions. This satisfied the reviewer.

4.	Top-k Accuracy Definition (n2SS): The reviewer pointed out the standard practice of using 2D InChIKeys for matching. The authors re-evaluated their results using this metric, discussed the implications, and agreed to add it to the final version for a fair comparison. This resolved the issue.